# Realizing four-electron conversion chemistry for all-solid-state Li‖I₂ batteries at room temperature

Zhu Cheng [1,2,3], Hang Liu[1,3], Menghang Zhang[1], Hui Pan [1], Chuanchao Sheng[1], Wei Li[1], Marnix Wagemaker [2], Ping He [1] ✉ & Haoshen Zhou [1] ✉

Rechargeable Li‖I₂ batteries based on liquid organic electrolytes suffer from pronounced polyiodides shuttling and safety concerns, which can be potentially tackled by the use of solid-state electrolytes. However, current all-solid-state Li‖I₂ batteries only demonstrate limited capacity based on a two-electron I⁻/I₂ polyiodides chemistry at elevated temperatures, preventing them from rivaling state-of-the-art lithium-ion batteries. Herein, we report a fast, stable and high-capacity four-electron solid-conversion I⁻/I₂/I⁺ chemistry in all-solid-state Li‖I₂ batteries at room temperature. Through the strategic use of a highly conductive, chlorine-rich solid electrolyte $Li_{4.2}InCl_{7.2}$ as the catholyte, we effectively activate the I₂/I⁺ redox couple. This activation is achieved through a robust I-Cl interhalogen interaction between I₂ and the catholyte, facilitated by an interface-mediated heterogeneous oxidation mechanism. Moreover, apart from serving as Li-ion conduction pathway, the $Li_{4.2}InCl_{7.2}$ catholyte is demonstrated to show a reversible redox behavior and contribute to the electrode capacity without compromising its conductivity. Based on the I⁻/I₂/I⁺ four-electron chemistry, the as-designed all-solid-state Li‖I₂ batteries deliver a high specific capacity of 449 mAh g⁻¹ at 44 mA g⁻¹ based on I₂ mass and an impressive cycling stability over 600 cycles with a capacity retention of 91% at 440 mA g⁻¹ and at 25 °C.

Rechargeable lithium-ion batteries, based on intercalation electrodes, have experienced significant advancements over the past few decades and continue to play a vital role in portable devices and electric vehicle markets today[1,2]. However, intercalation electrodes are approaching their maximum capacity limitations since their structures can only accommodate a maximum of one lithium ion per formula unit for insertion and release. Even the extensively studied Li-rich electrodes can only incorporate less than two lithium ions per formula unit[3,4], limiting the specific capacities achievable. Furthermore, intercalation electrodes typically utilize transition metal elements like cobalt and nickel, which are not only expensive but also environmentally detrimental[5]. Hence, it is crucial to discover and design cost-effective electrode materials capable of storing a greater number of lithium ions, enabling multi-electron reactions. This advancement is essential for achieving higher specific capacity and energy density in future energy storage systems.

As an emerging conversion electrode material, iodine (I₂) shows several fascinating advantages including high abundance, environmentally benign, and a satisfactory specific capacity of 211 mAh g⁻¹ derived from two-electron I⁻/I₂ redox reaction[6,7]. Specifically, typical

¹Center of Energy Storage Materials & Technology, College of Engineering and Applied Sciences, Jiangsu Key Laboratory of Artificial Functional Materials, National Laboratory of Solid State Microstructures and Collaborative Innovation Center of Advanced Microstructures, Nanjing University, Nanjing, PR China. ²Section Storage of Electrochemical Energy, Radiation Science and Technology, Faculty of Applied Sciences, Delft University of Technology, Delft, The Netherlands. ³These authors contributed equally: Zhu Cheng, Hang Liu. ✉e-mail: pinghe@nju.edu.cn; hszhou@nju.edu.cn

liquid Li‖I₂ batteries exhibit good rate capabilities attributed to the high solubility and rapid diffusion of polyiodides in organic liquid electrolytes. However, this advantage comes with a critical drawback in the form of polyiodide shuttling, which continuously consumes the Li inventory at the negative electrode, ultimately leading to the failure of the battery[8,9] (Fig. 1). Replacing liquid electrolytes with solid-state electrolytes could perfectly avoid the polyiodide shuttle effect because the solid-state electrolytes only allow lithium ions to pass through. Simultaneously, the potential safety problem associated with liquid electrolytes can be well resolved by the use of inflammable and thermal stable solid-state electrolytes[10,11]. By employing a hybrid solid electrolyte, recently our group has realized a "confined dissolution" polyiodides chemistry that enables long-life rechargeable all-solid-state (ASS) Li‖I₂ batteries[12]. Unfortunately, this battery can only show a satisfactory capacity of 200 mAh g⁻¹ at elevated temperature, severely restricting its potential application scenarios. To compete with state-of-the-art lithium-ion batteries, ASS Li‖I₂ batteries need to demonstrate the ability to operate at room temperature (RT), and more importantly, achieve higher capacity and energy density.

In conventional Li‖I₂ batteries, the redox of I₂ electrode is realized via a two-electron I⁻/I₂ process. In fact, iodine possesses multiple valence states (such as I⁻, I₂, I⁺, I³⁺, and I⁵⁺) and thus is capable of exhibiting multi-electron redox behavior[13,14]. Recently Zhi's group[15] and Liang's group[16] reported the activation of I₂/I⁺ redox couple in aqueous Zn‖I₂ batteries. This is achieved by the utilization of high Cl⁻ concentration electrolytes, which offer a rich coordination environment for the formation of I−Cl interhalogen bonds. Based on the pioneering work in liquid system, we can boldly speculate that designing ASS Li‖I₂ batteries with a similar four-electron chemistry could potentially lead to the simultaneous achievement of high-energy density and enhanced safety features.

Unlike in aqueous Zn‖I₂ batteries, where the polyiodide species and Cl⁻ can permeate the entire I₂ electrode to form robust I−Cl interhalogen bonds easily, in ASS Li‖I₂ batteries, both I₂ and the solid electrolyte are fixed at their local environments due to the immobile nature of solids. Consequently, in order to fully activate the I₂/I⁺ redox in the solid system, it is necessary to establish sufficient interhalogen coordination environment near every I₂ particle. In addition, the strong oxidation property of I⁺ also requires the solid electrolyte to possess high oxidation tolerance. Following this guideline, in this work we develop a four-electron ASS Li‖I₂ battery by designing a I₂/Li₄.₂InCl₇.₂ (LIC7.2) nanocomposite electrode. Compared with normal Li₃InCl₆ (LIC6), LIC7.2 could offer a more abundant Cl coordination environment while retaining high-voltage stability and sufficient ionic

conductivity. Consequently, we anticipate a more robust I₂/I⁺ redox behavior in the I₂/LIC7.2 electrode compared to the I₂/LIC6 electrode. On top of this, a reversible I⁻/I₂/I⁺ four-electron solid-phase conversion chemistry is successfully achieved, as unveiled by a series of Raman, X-ray photonic spectroscopy (XPS) measurements and density function theory (DFT) calculations. This leads to a high coulombic efficiency close to 100% and a doubled specific capacity for the ASS Li‖I₂ batteries (Fig. 1). Moreover, dissimilar to conventional ASS batteries where the catholyte merely acts as a fast ion transport channel, here LIC7.2 is demonstrated to offer a reversible capacity without compromising its convenient ion transport and stability over a voltage range of 2−4 V versus Li⁺/Li. As a result, the designed ASS Li‖I₂ batteries show long cycle life at a practical high areal capacity (1.42 mAh cm⁻²) and at RT (specified as 25 °C unless otherwise noted in the following section). The superior performance is also demonstrated at elevated temperature (60 °C) and high areal capacity (6.75 mAh cm⁻²) scenario. Furthermore, the as-fabricated ASS Li‖I₂ pouch cell shows reliable safety characteristic in cut tests. This work enriches the fundamental understanding of solid-phase reaction chemistry of I₂ electrode and will inspire more researches to explore novel ASS battery systems with high-energy densities.

## Results and discussion
### Activation of I₂/I⁺ redox in ASS Li‖I₂ batteries
The key to whether iodine molecules can undergo a four-electron reaction in an ASS system lies in the oxidation of zero-valent iodine to positively charged iodine. The local environment in the vicinity of the I₂ particles, as provided by the catholyte, plays a crucial role in this process. Building upon previous research on aqueous chlorine-concentrated Zn‖I₂ batteries, here we utilize halide solid electrolytes as the catholyte in an endeavor to trigger the I₂/I⁺ redox reaction through interhalogen bonds. Chloride LIC6 and bromide Li₃YBr₆ (LYB) are first considered as the catholyte candidates. Prior to mixing them with I₂ particles, charge tests were conducted on electrodes without iodine, only with electrolyte (LIC6 or LYB) and conductive agent carbon, to observe the oxidation behavior of the electrolytes themselves. The LIC6 and LYB exhibit capacities of 29.5 mAh g⁻¹ and 14.4 mAh g⁻¹, respectively (Fig. 2a), attributed to the oxidation of the halide elements[17], since In and Y element are already at their highest oxidation state of +3. Then nanocomposite I₂ electrode was prepared by mixing the LIC6 or LYB with I₂ and carbon via high-energy ball milling (denoted as I₂/LIC6 and I₂/LYB, respectively). As shown in Fig. 2b, the I₂/LYB electrode exhibits a negligible charging capacity,

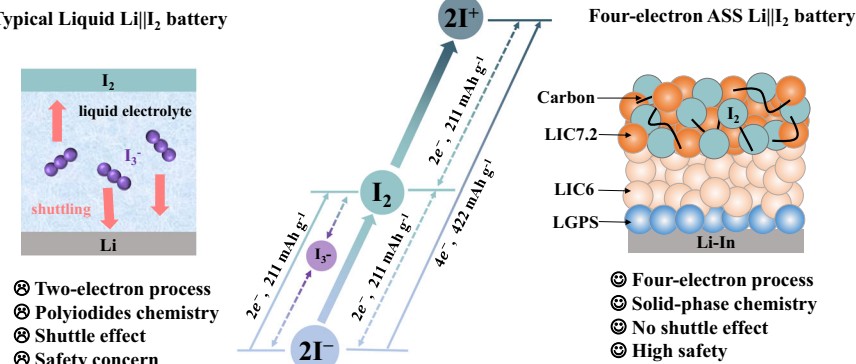

**Typical Liquid Li‖I₂ battery**

⊗ Two-electron process
⊗ Polyiodides chemistry
⊗ Shuttle effect
⊗ Safety concern

**Four-electron ASS Li‖I₂ battery**

☺ Four-electron process
☺ Solid-phase chemistry
☺ No shuttle effect
☺ High safety

**Fig. 1 | Schematic representations of liquid Li‖I₂ batteries and LIC7.2-based ASS Li‖I₂ batteries.** Typical liquid Li‖I₂ batteries show a theoretical capacity of 211 mAh g⁻¹ based on two-electron I⁻/I₃⁻/I₂ polyiodides chemistry. The use of liquid electrolytes which dissolve polyiodides leads to severe shuttle effect and thus a low coulombic efficiency, as well as the potential safety concern. In the proposed ASS Li‖I₂ batteries, a fast and reversible four-electron solid-phase conversion chemistry is achieved by the design of I₂/LIC7.2 composite electrode with a stable chlorine-rich environment, which

is capable of activating the I₂/I⁺ redox couple. A doubled theoretical capacity of 422 mAh g⁻¹ and high specific energy of ~1302 Wh kg⁻¹ based on the active I₂ mass (434 Wh kg⁻¹ based on the total mass of I₂ and LIC7.2) are demonstrated for the ASS Li‖I₂ batteries. The use of LIC7.2 electrolytes perfectly eliminates the polyiodides shuttle problem and improves the safety. The crystalline LIC6 is used as the solid electrolyte layer due to its high ionic conductivity, and a layer of Li₁₀GeP₂S₁₂ (LGPS) is adopted between LIC6 and Li−In electrode to prevent the reduction of LIC6.

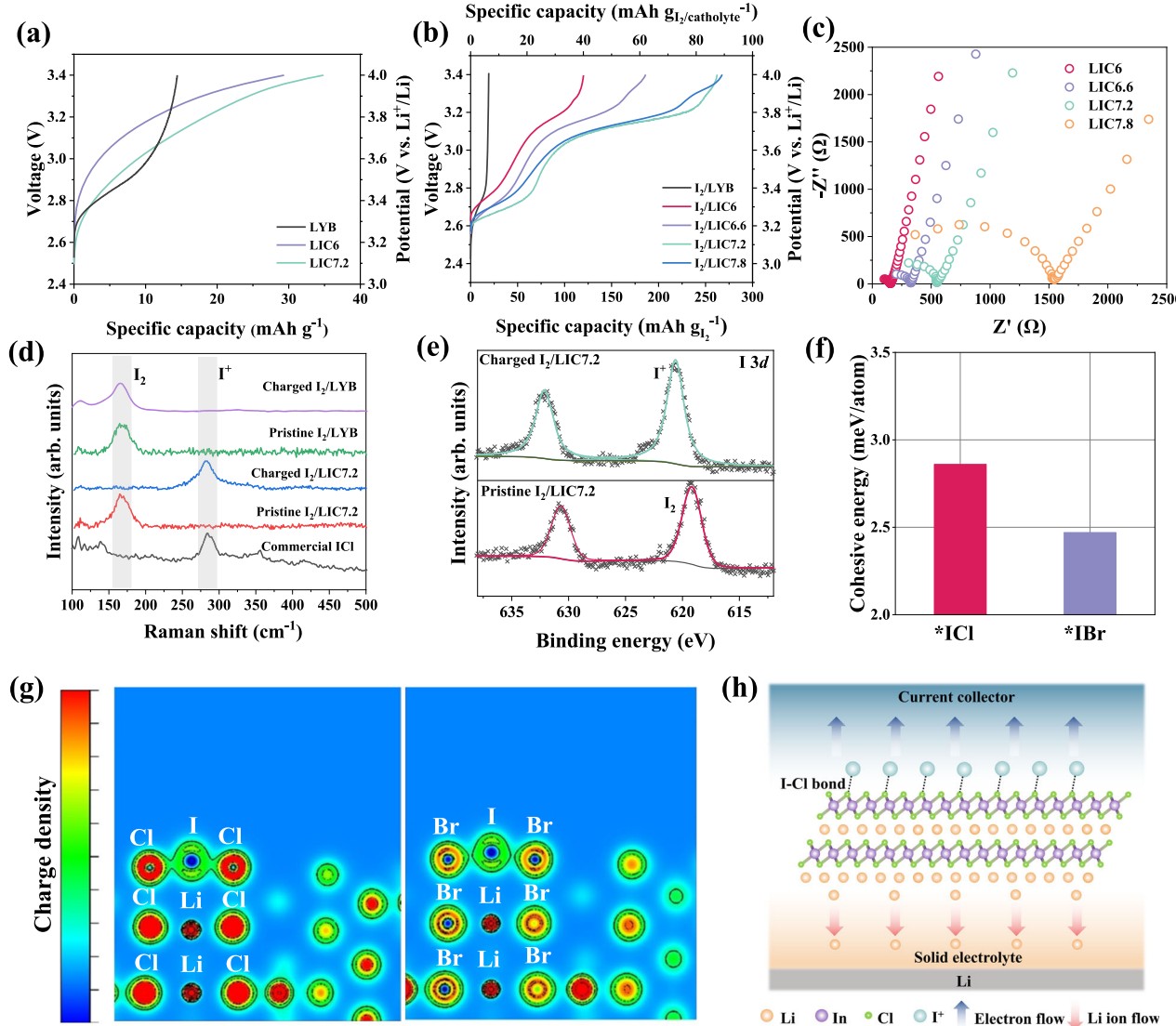

**Fig. 2 | Electrochemical behavior of I$_2$ electrode in ASS batteries based on different electrolytes. a** Direct charging voltage profiles of LYB, LIC6, and LIC7.2 as electrode materials. The specific capacity is calculated based on the mass of LYB, LIC6, or LIC7.2. **b** Direct charging voltage profiles of the I$_2$/LYB, I$_2$/LIC6, I$_2$/LIC6.6, I$_2$/LIC7.2, and I$_2$/LIC7.8 electrode. The specific capacities based on the I$_2$ mass and the total mass of I$_2$ and catholyte are displayed in bottom and top x-axis, respectively. **c** Nyquist plots of the prepared LIC6, LIC6.6, LIC7.2, and LIC7.8. **d** Raman spectra of the commercial ICl compound, pristine/charged I$_2$/LYB and I$_2$/LIC7.2 electrodes. **e** I 3$d$ XPS spectra of the pristine and charged I$_2$/LIC7.2 electrode. **f** Two-dimensional charge densities distribution of *IBr in I$_2$/LYB and *ICl in I$_2$/LIC7.2. **g** Cohesive energies of the *IBr in I$_2$/LYB and *ICl in I$_2$/LIC7.2 obtained from DFT calculations. **h** Schematic of the interface-mediated heterogeneous oxidation mechanism facilitated by the substantial nanointerface between I$_2$ and LIC7.2 during the charging process of I$_2$/LIC7.2 electrode.

indicating that I$_2$ cannot undergo effective oxidation in the bromide-based electrolyte. On the contrary, a higher capacity of 120 mAh g$^{-1}$ is attained for the I$_2$/LIC6 electrode. The charging plateau at 3.3 V versus Li$^+$/Li with a capacity around 60 mAh g$^{-1}$ is attributed to the conversion of reduced polyiodides species to I$_2$. During the ball-milling process, I$_2$, with its strong electron absorption capability, induces electron transfer from KB to I$_2$ and therefore leads to the formation of reduced polyiodides species such as I$_5^-$ and I$_3^{-}$[12,18]. As the conversion from I$_5^-$ to I$_2$ has a theoretical capacity of 42.2 mAh g$^{-1}$, there is most likely a small amount of other further reduced polyiodides species (I$_3^-$) existing in the pristine electrode. The second charging plateau at 3.75 V versus Li$^+$/Li likely results from the oxidation of I$_2$. The oxidation of Cl$^-$, which typically occurs at a higher potential, appears to be suppressed here, possibly due to a strong I–Cl interaction. However, this capacity is significantly lower than the theoretical capacity (211 mAh g$^{-1}$) of the two-electron I$_2$/I$^+$ conversion. This indicates that while the chloride solid electrolyte LIC6 has the potential to activate the oxidation of I$_2$, it cannot induce the complete I$_2$/I$^+$ redox couple. This limitation is likely due to the insufficient chlorine concentration surrounding the I$_2$ particles.

To enhance the localized chlorine concentration around the I$_2$ particles, it is necessary to increase the chlorine content in the LIC6 catholyte. At the same time, a high ionic conductivity for the catholyte is also required to provide efficient Li-ion transport within the I$_2$ electrode. Therefore, a series of Cl-concentrated Li–In–Cl solid electrolytes, namely Li$_{3.6}$InCl$_{6.6}$ (LIC6.6), Li$_{4.2}$InCl$_{7.2}$ (LIC7.2), and Li$_{4.8}$InCl$_{7.8}$ (LIC7.8) were synthesized by one-step ball-milling process. Supplementary Fig. 1 shows the X-ray diffraction (XRD) patterns of these Li–In–Cl series. All of them can be indexed with a C2/m space group, while some minor peaks corresponding to LiCl are shown in the XRD pattern of LIC7.8. This implies that the LIC7.8 stoichiometry is out of the Li–In–Cl solid solution range. Electrochemical impedance spectroscopy (EIS) measurements were conducted to obtain the ionic conductivities of the above-prepared Li–In–Cl solid electrolyte. With

increasing Cl content, the ionic conductivity drops from 0.63 mS/cm for LIC6 to 0.06 mS/cm for LIC7.8 (Fig. 2c). To evaluate their capabilities to activate the $I_2/I^+$ redox couple, $I_2$ composite electrodes based on these catholytes were subjected to direct charging tests. As displayed in Fig. 2b, the charge capacities for $I_2$/LIC6, $I_2$/LIC6.6, $I_2$/LIC7.2, and $I_2$/LIC7.8 electrodes are 120, 186, 261, and 267 mAh g$^{-1}$, respectively. It is clear that enough high Cl content in the Li–In–Cl catholyte (Cl ≥ 7.2 in Li–In–Cl) is required to fully activate the $I_2/I^+$ redox reaction. As a result, to achieve both efficient ionic transport and full utilization of $I_2/I^+$ redox couple, the LIC7.2 with high Cl content and a satisfactory ionic conductivity of 0.18 mS cm$^{-1}$ is selected as the optimized catholyte in this system. The Rietveld refinement results (Supplementary Fig. 2 and Tables S1 and S2) and $^6$Li solid-state magic angle spinning nuclear magnetic resonance measurements (Supplementary Fig. 3) further demonstrate that the stoichiometry LIC7.2 is a pure phase without any crystalline or amorphous unreacted precursors. Elemental mapping (Supplementary Fig. 4 and Table S3) also corroborated the expected stoichiometry of the prepared LIC7.2. It should be noted that the LIC7.2 alone can only exhibit a charge capacity of 35 mAh g$^{-1}$ (Fig. 2a), whereas the $I_2$/LIC7.2 electrode exhibits a much higher specific capacity of 265 mAh g$^{-1}$ (Fig. 2b). The plateau at 3.75 V versus Li$^+$/Li, corresponding to the oxidation of $I_2$ molecules, accounts for a capacity of 190 mAh g$^{-1}$, which is very close to the theoretical capacity of the $I_2/I^+$ redox couple.

Ex situ Raman measurements were performed to get insight into the evolution of $I_2$ species in the composite $I_2$ electrode. As a comparison, the Raman spectra of both $I_2$/LYB and $I_2$/LIC7.2 electrode at pristine and charged states are displayed in Fig. 2d. Clear Raman shift at ~162 cm$^{-1}$ of $I_5^-$, as the major reduced $I_2$ species, is observed for pristine $I_2$/LYB and $I_2$/LIC7.2 electrodes. After direct charging, no difference is found in the Raman spectrum of the $I_2$/LYB electrode, proving that $I_2$ cannot be oxidized with the LYB catholyte. In contrast, the $I_5^-$ signal disappears and a new Raman shift located at ~280 cm$^{-1}$ emerges for the charged $I_2$/LIC7.2 electrode. This Raman signal is also evident for commercial ICl compound, indicating the existence of the interhalogen bonds in the charged $I_2$/LIC7.2 electrode. This is consistent with the solid-state UV-Vis spectroscopy results (Supplementary Fig. 5), which show a strong absorption peak at around 350 nm—a characteristic peak of ICl interhalogen formation[16] for the charged $I_2$/LIC7.2. Further confirmation of this $I_2/I^+$ conversion comes from the XPS results, showing that the $I_2$ (619.5 eV) in the pristine $I_2$/LIC7.2 electrode is oxidized to I$^+$ (620.5 eV) after being charged to 4 V versus Li$^+$/Li (Fig. 2e). Based on above results, it can be inferred that the interhalogen bond between Cl$^-$ and I$^+$ is more robust than that between Br$^-$ and I$^+$. This explains why the $I_2/I^+$ conversion is not activated for the $I_2$/LYB electrode.

To better understand the important role of chlorine in activating $I_2/I^+$ redox couple, DFT calculations were conducted on the $I_2$/LIC7.2 and $I_2$/LYB substrates (Supplementary Fig. 6). *ICl (*IBr) represents the interhalogen interaction between LIC7.2 (LYB) and I$^+$ species. A higher cohesive energy value is obtained for $I_2$/LIC7.2 than $I_2$/LYB, proving that the interhalogen interaction between $I_2$ and LIC7.2 is more thermodynamically favorable than that between $I_2$ and LYB (Fig. 2f). The related electron localization function was carried out to analyze the distribution of lone pair electron of the interhalogen phases. As shown in Fig. 2g, much more charges are found to distribute between I–Cl bonds in $I_2$/LIC7.2 than I–Br bonds in $I_2$/LYB, reflecting a more stable and robust electrical coupling between LIC7.2 and I$^+$. This is also consistent with the Bader charge calculations showing that more charges are transferred between I and Cl (Supplementary Fig. 6a). Besides, the calculated bond length is shorter for I–Cl in $I_2$/LIC7.2 than I–Br in $I_2$/LYB (Supplementary Fig. 6b), indicating a more robust binding and stronger interaction between LIC7.2 and I$^+$ species. This suggests a lower energy barrier for the $I_2/I^+$ conversion in a LIC7.2-based system.

During the charging process, oxidation and Li-ion extraction typically occur within the same phase for intercalation electrodes, such as LiCoO$_2$. These electrodes generally exhibit sufficient Li-ion diffusivity, allowing the delithiation process to occur easily in bulk form. However, for conversion electrodes with poor Li-ion mobility, the achievable capacity is highly dependent on their particle size. Smaller particle sizes lead to substantial interfaces between particles for conversion reaction to take place thus maximizing capacity release. This interface-mediated heterogeneous oxidation mechanism has been reported and widely acknowledged for conversion-type metal oxide electrode materials[19–21]. For example, for the Co/Li$_2$O composite electrode, when the material's size is at the micron scale, its capacity is quite limited, and only at the nanoscale can it achieve a significantly large specific capacity during charging (over 600 mAh/g)[20]. During charging, nanoscale Co undergoes an interface-mediated heterogeneous oxidation induced by the interface with Li$_2$O. As Li$_2$O releases lithium ions, oxygen ions bind to the Co atoms on the surface of the metallic Co, with these Co atoms simultaneously losing electrons, forming Co$_3$O$_4$. Both metallic Co and the charging product Co$_3$O$_4$ have poor ionic conductivity for oxygen ions. However, Co/Li$_2$O can still achieve a high charging capacity and a considerable amount of Co$_3$O$_4$ conversion. This is the result of interfacial-induced heterogeneous oxidation reactions at the nanoscale. A similar reaction occurs in the $I_2$/LIC7.2 electrode as shown in Fig. 2h and Supplementary Fig. 7. Nanoscale iodine and lithium halide electrolyte also form a pair of charging reactants. During the charging process, the lithium halide electrolyte (LIC7.2) releases lithium ions, and the halide ions bind to the iodine on the surface of the iodine particles. At the same time, the iodine molecules lose electrons and bond with Cl ions to form interhalogen. Of course, the halide electrolyte can only conduct lithium ions, but at the nanoscale, $I_2$/LIC7.2 electrode can still achieve a high conversion rate and large capacity by interface-mediated heterogeneous oxidation reaction. It should be pointed out that the difference between Co/Li$_2$O and $I_2$/LIC7.2 systems is that, Co atoms ionically bond with O atoms after charging, while the activated I ions covalently bond with Cl ions. This is because (1) I and Cl are both nonmetals and form a bond by sharing electron pairs, (2) the difference in electronegativity between I and Cl is not large enough to create an ionic bond. To achieve a high conversion efficiency as possible for the $I_2/I^+$ redox reaction, super intense ball-milling process (500 rpm for 12 h in WC jars) is employed to create substantial nanointerface in the composite $I_2$/LIC7.2 electrode. As a result, a large amount of $I_2$ and LIC7.2 domains with a size of several nanometers are created during this ball-milling process, as seen in the transmission electron microscopy image in Supplementary Fig. 8a. This nanoscale homogenous mixing is further demonstrated by energy-dispersive X-ray mapping shown in Supplementary Fig. 8b. For comparison, a hand-ground $I_2$/LIC7.2 electrode with inadequate contact interfaces is also tested, resulting in a significantly lower capacity of just 23 mAh g$^{-1}$ (Supplementary Fig. 9). This further proves the interface-mediated reaction mechanism for $I_2/I^+$ conversion and emphasizes the critical role of electrode preparation methods in achieving high conversion capacity for the $I_2/I^+$ reaction.

## Four-electron conversion reaction mechanism

The activation of the two-electron $I_2/I^+$ process has been successfully demonstrated for the $I_2$/LIC7.2 electrode. However, it is still questionable whether a reversible $I_2/I^+$ process and the whole four-electron reaction can be realized. To investigate this, we conducted discharge/charge experiments on the LIC7.2-based ASS Li||$I_2$ battery, focusing on the first cycle (a two-electron process) and the second cycle (a four-electron process). As $I_2$ is the middle valence state between I$^-$ and I$^+$, the $I_2$/LIC7.2 electrode can be charged first to I$^+$ or discharged first to I$^-$. The battery is first discharged to 2 V versus Li$^+$/Li here and then recharged to 4 V versus Li$^+$/Li (Supplementary Fig. 10a). Note that the capacities are based on active $I_2$ mass in the $I_2$/LIC7.2 electrode. During the first discharge, a capacity of 242 mAh g$^{-1}$ is attained, most likely

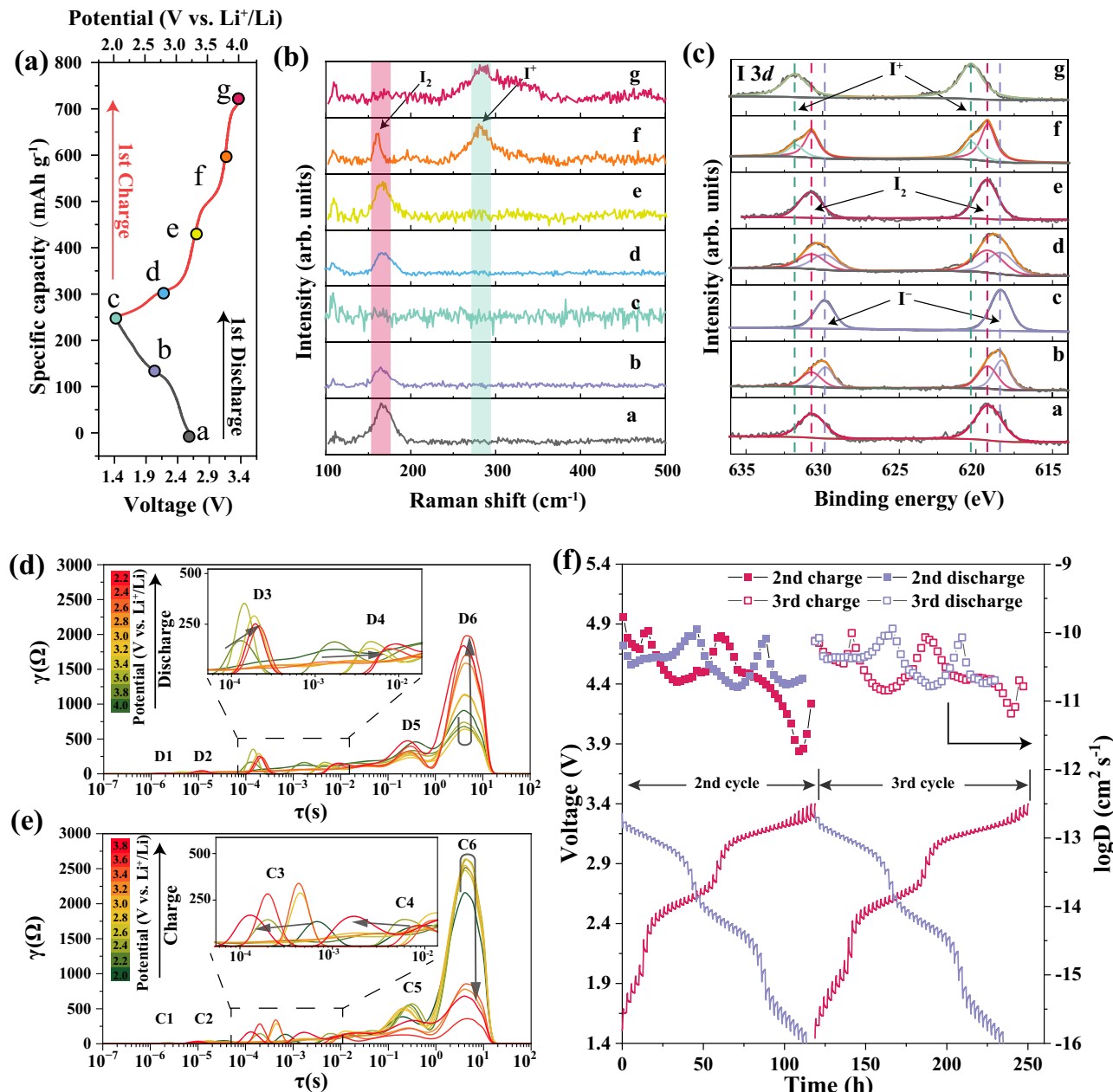

**Fig. 3 | Characterizations of the reaction mechanism and kinetics for LIC7.2-based ASS Li∥I₂ batteries. a** First cycle discharge/charge curve of the I₂/LIC7.2 electrode. The specific capacity is calculated based on the mass of I₂. **b** Ex situ Raman spectra of the I₂/LIC7.2 electrode at stages a–g corresponding to the stages shown in (**a**). **c** Ex situ XPS spectra of the I₂/LIC7.2 electrode at stages a–g corresponding to the stages shown in (**a**). **d** DRT analysis of the discharge process for Li∥I₂/LIC7.2 battery at the second cycle. **e** DRT analysis of the charging process for the Li∥I₂/LIC7.2 battery at the second cycle. **f** GITT curves and the corresponding calculated diffusion coefficients of the Li∥I₂/LIC7.2 battery at the second and third cycles. Each constant current-rest step consisted of a 0.02 mA cm⁻² current pulse of 30 min and a 2 h relaxation. For these ex situ characterizations, the employed I₂/LIC7.2 electrode with 25% of active I₂ and an I₂ mass loading of 2 mg cm⁻² is electrochemically tested at 22 mA g⁻¹ with a voltage window of 2–4 V vs. Li at 25 °C.

corresponding to the transformation from I₂ to I⁻. During the subsequent recharge process, the battery displays three distinct plateaus (which will be discussed in the next section), with a noteworthy doubling of capacity observed at around 477 mAh g⁻¹. A similar discharge/charge behavior and a high capacity of around 420 mAh g⁻¹ is observed for the battery at the second cycle. When the battery is first charged to 4 V versus Li⁺/Li and then discharged to 2 V versus Li⁺/Li for the first cycle, it still shows a similar voltage profile and capacity value for the second cycle (Supplementary Fig. 10b), providing compelling evidence for the highly reversible nature of the I₂ redox reaction in this system.

As the crystallinity of iodine gets destroyed during high-energy ball-milling process (Supplementary Fig. 11), XRD was unable to distinguish various I₂ species at different states of charge (SOC). To unveil the detailed reaction mechanism of I₂/LIC7.2 electrode in the four-electron conversion ASS Li∥I₂ battery, both ex situ Raman and XPS measurements were carried out on I₂/LIC7.2 electrode at different SOC to monitor the evolution of I₂ species. The SOCs for Raman and XPS measurements are marked in the discharge-charge profile in Fig. 3a. At open circuit voltage, stage a in Fig. 3b, a strong Raman signal shows at ~160 cm⁻¹, which becomes weaker at a deeper discharge state of 2.6 V versus Li⁺/Li (stage b) and could be hardly observed after full discharge

to 2 V versus Li$^+$/Li because of the insensitivity of I$^-$ to Raman measurements (stage c). This indicates the transformation from I$_2$ to I$^-$ (2e$^-$) during the discharge process. In the following charge process, the Raman signal located at ~160 cm$^{-1}$ appears again when the battery is charged to 2.8 V versus Li$^+$/Li (stage d) and continues to grow after charging to 3.25 V versus Li$^+$/Li (stage e), corresponding to the oxidation process from I$^-$ to I$_2$. The I$_2$ signal intensity decreases when the electrode is further charged to 3.7 V versus Li$^+$/Li (stage f). In addition, a strong stretch of I–Cl bond starts to emerge at ~280 cm$^{-1}$, proving that the I$_2$ is successfully oxidized to I$^+$. A full I$_2$/I$^+$ conversion is achieved after the electrode is charged to 4 V versus Li$^+$/Li (stage g), accompanied by the disappearance of I$_2$ Raman signal. It should be pointed out that this whole four-electron conversion process does not involve the dissolution-state polyiodides since the Raman shift at ~116 cm$^{-1}$ corresponding to I$_3^-$ is not observed during the measurements, which are commonly seen in previous liquid[22] or polymer-based[12] Li‖I$_2$ batteries. This is the first time that a reversible solid-phase conversion was demonstrated in ASS Li‖I$_2$ batteries, and therefore, no detrimental shuttle effect is expected.

The evolution of valence state change of I$_2$ species is precisely probed by XPS measurements at exactly the same SOCs with Raman measurements (Fig. 3c). From stage a to stage c which corresponds to the discharge process, the strong peak located at ~619 eV (I$_2$) is found to decrease and then vanish, while the XPS signal peak at ~618 eV (I$^-$) grows with a deeper states of discharge (SOD). In the following charge process (stage d–g), XPS signal of I$^-$ disappears at a charge state of 3.25 V versus Li$^+$/Li (stage e) and I$_2$ signal becomes stronger, corresponding to the I$^-$/I$_2$ conversion at 2–3.25 V versus Li$^+$/Li. After charging to stage f, besides the XPS signal of I$_2$, another dominant peak can be observed at ~620.5 eV, indicating that I$_2$ is oxidized to I$^+$ under the inducing effect of LIC7.2. As expected, only I$^+$ peak at ~620.5 eV is detected at the charge cutoff voltage of 4 V versus Li$^+$/Li (stage g).

To verify the fast and reversible kinetics in I$_2$/LIC7.2-based ASS Li‖I$_2$ batteries with a four-electron chemistry, the battery resistance evolution was further studied by the in situ EIS measurements (Supplementary Fig. 12). The distribution of relaxation times (DRT) analysis was conducted at various SOD and SOC in order to provide a more intuitive representation of how the resistance of each component of the battery evolves throughout the cycling processes (Fig. 3d, e). Six dominant peaks can be identified in the time scale shown, which correspond to different kinetic processes with distinct time constants. During the whole discharge/charging process, the D1/C1 peak ranging between 10$^{-6}$ s and 10$^{-5}$ s representing the grain boundary resistance of the solid electrolyte[23] is almost submerged by the noise or inductance signal at super high frequencies. The D2/C2 peak located around 10$^{-5}$ s corresponds to the contact resistance between electrode and current collectors. The peaks with the intermediate processes are attributed to the charge transfer behavior at the LIC6/Li$_{10}$GeP$_2$S$_{12}$ (LGPS) interface (D3/C3, LIC6, and LGPS are used as a dual-layer electrolyte here), Li–In negative electrode/LGPS interface (D4/C4) and I$_2$/LIC7.2 electrode/LIC6 interface (D5/C5). As the solid-state diffusion within the I$_2$/LIC7.2 electrode is the most sluggish process, it shows the largest time constant above 10$^0$ s and is assigned to the D6/C6 peak. During the discharge process, the D6 peak shows a decrease during the first several measurements where the transformation from I$^+$ to I$_2$ is anticipated. Afterward, an increase in intensity is observed for D6 along the following discharge process, indicating a slower diffusion within the I$_2$/LIC7.2 electrode. In the subsequent charge process, the intensity of C6 peak slightly increases at first and then decreases until the battery is fully charged. Similarly, reversible charge transfer processes at the interfaces are also observed. Time constants of the charge transfer processes at LIC6/LGPS and Li–In negative electrode/LGPS interfaces shift to a larger value during discharge (D3 and D4 peak) and then shift back to their original states during charge. Comparing the beginning of discharge and end of charge states, all the C1–C6 peaks show almost

the same intensities and shifts as D1–D6 peaks, implying that the solid-phase four-electron chemistry is highly reversible. This is further verified by the galvanostatic intermittent titration technique (GITT) measurement (Fig. 3f). The GITT experiment was carried out on the battery during the second and the third cycles, with each step consisting of a 30-min charge/discharge at a constant current of 0.02 mA cm$^{-2}$ and a 2 h rest process. Lithium diffusion coefficients (D$_{Li}^+$) during these two cycles are calculated from the GITT measurements. At the initial states of both I$^-$/I$_2$ and I$_2$/I$^+$ solid-phase reactions, D$_{Li}^+$ values are relatively high and then gradually decrease afterward. During the whole two cycles, the D$_{Li}^+$ is in the range of 10$^{-11}$ and 10$^{-10}$ cm$^2$ s$^{-1}$ except at the end of charge, indicating a fast kinetics of this I$^-$/I$_2$/I$^+$ solid-phase conversion chemistry. The evolution trend and value of D$_{Li}^+$ are nearly the same for the second and the third cycle, which again demonstrates the reversibility of this four-electron I$^-$/I$_2$/I$^+$ conversion mechanism.

## Capacity contribution of LIC7.2 catholyte

As mentioned above, three discharge/charge plateaus are observed in the voltage profile of I$_2$/LIC7.2 electrode. Similarly, three pairs of redox peaks could also be well defined in its cyclic voltammetry (CV) curves within the operation voltage window of 2–4 V versus Li$^+$/Li (Fig. 4a). The cathodic peak at ~2.75 V versus Li$^+$/Li ($E_{c,2}$) and anodic peak at ~3.3 V ($E_{a,2}$) are rationally ascribed to the reversible reaction between I$_2$ and I$^-$, while the pair of peaks of $E_{c,3}$ (~3.6 V versus Li$^+$/Li) and $E_{a,3}$ (~3.75 V versus Li$^+$/Li) is attributed to the I$_2$/I$^+$ redox couple. According to previous DFT studies[24,25], the indium-based halide electrolyte would be reduced at voltages lower than 2.3 V versus Li$^+$/Li, which falls within the cycling voltage range of the ASS Li‖I$_2$ battery (2–4 V versus Li$^+$/Li). Therefore, the third pair of redox peaks ($E_{c,1}$ and $E_{a,1}$) centered at around 2.3 V versus Li$^+$/Li is most likely attributed to the In$^{3+}$/In$^{2+}$ redox couple in LIC7.2 catholyte.

To clarify the redox activity of LIC7.2 catholyte in the I$_2$/LIC7.2 composite electrode, LIC7.2 mixed with KB was used as the positive electrode to perform CV tests. Within a voltage range of 2–4 V versus Li$^+$/Li, a cathodic peak $E_{c,1}'$ (located at ~2 V versus Li$^+$/Li) and two anodic peaks $E_{a,1}'$ and $E_{a,4}$ (located at 2.75 V versus Li$^+$/Li and 4 V versus Li$^+$/Li, respectively) can be observed (Fig. 4b). When the CV scans are conducted at 2.4–4 V versus Li$^+$/Li, as seen in Supplementary Fig. 13, no obvious cathodic peaks can be distinguished and $E_{a,1}'$ anodic peak disappear as well. Therefore, both $E_{c,1}'$ and $E_{a,1}'$ come from the redox reaction of In$^{3+}$/In$^{2+}$, while $E_{a,4}$ is attributed to the oxidation of Cl$^-$ according to previous work[26]. Importantly, $E_{c,1}'/E_{a,1}'$ redox pair of LIC7.2 electrode occurs at a similar potential to $E_{c,1}/E_{a,1}$ redox pair of I$_2$/LIC7.2 electrode, proving that the $E_{a,1}$ and $E_{c,1}$ peaks in the I$_2$/LIC7.2 electrode indeed come from the LIC7.2 catholyte. It should be pointed out that the $E_{a,4}$ peak corresponding to the Cl$^-$ oxidation in LIC7.2 disappears in the CV curves of I$_2$/LIC7.2 electrode, indicating that Cl$^-$ redox behavior of LIC7.2 is different from that of I$_2$/LIC7.2 electrode. This could be explained by the robust I–Cl interhalogen interaction between activated I$^+$ and Cl$^-$, which suppresses the oxidation of Cl$^-$ within this voltage range.

The capacity contribution of LIC7.2 redox activity to the I$_2$/LIC7.2 electrode capacity was quantified by galvanostatic cycling test of the LIC7.2 electrode following a charge-discharge sequence. As shown in Fig. 4c, during the first charge, the LIC7.2 delivers a capacity of 27 mAh g$^{-1}$ derived from the oxidation of Cl$^-$. The subsequent discharge process reveals a capacity of 62 mAh g$^{-1}$ mainly attributed to the reduction of In$^{3+}$. However, by the second cycle, the charging plateau associated with Cl$^-$ almost disappears, implying that the Cl$^-$ redox reaction of LIC7.2 is not reversible. Furthermore, the discharging capacity linked to the In$^{3+}$/In$^{2+}$ redox initially decreases but stabilizes by the tenth cycle. Notably, a stable capacity of 25 mAh g$^{-1}$ is observed for the LIC7.2 electrode after 10 cycles (Fig. 4d). Given that the mass ratio of LIC7.2 to I$_2$ is 2:1 in the I$_2$/LIC7.2 electrode, it is evident

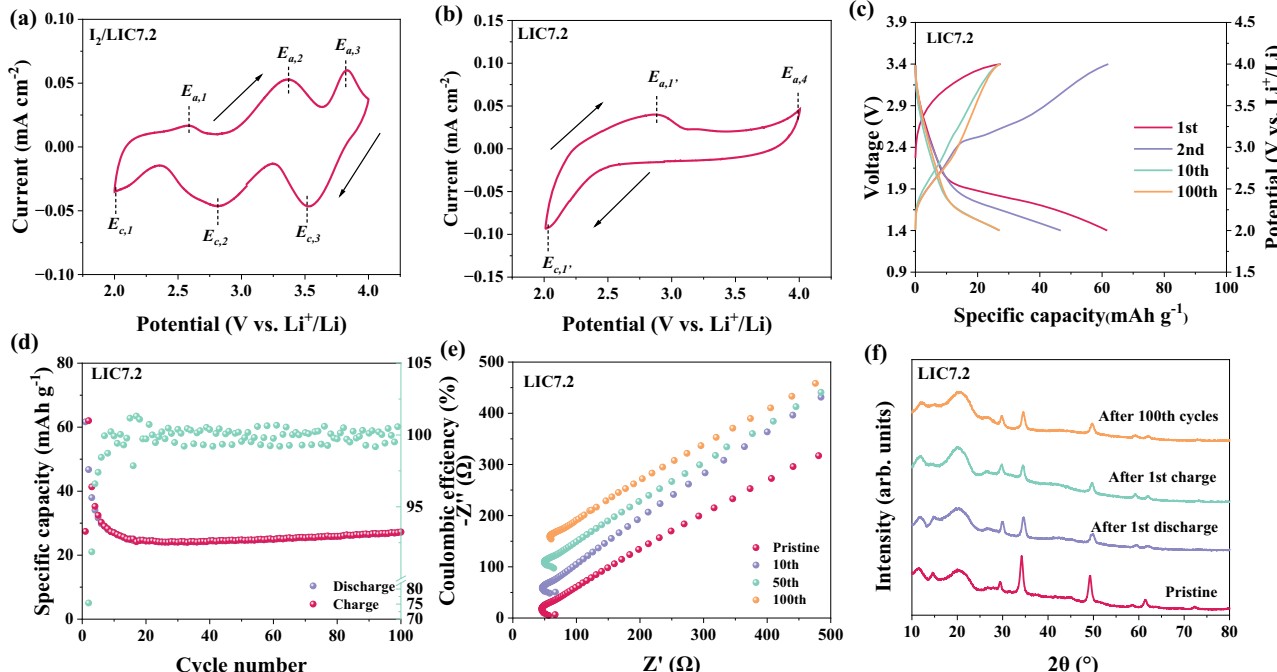

**Fig. 4 | Redox activity of LIC7.2. a** Cyclic voltammetry curves of the $I_2$/LIC7.2 electrode (The $I_2$ mass loading is 0.5 mg cm$^{-2}$) at a sweep rate of 0.1 mV s$^{-1}$. **b** CV curves of the LIC7.2 electrode (The LIC7.2 mass loading is 2 mg cm$^{-2}$) with a sweeping voltage range of 2–4 V versus Li$^+$/Li at a sweeping rate of 0.1 mV s$^{-1}$. **c** Voltage profiles of the LIC7.2 electrode at the 1st, 2nd, 10th, and 100th cycle. The electrode was charged first and then discharged. The specific capacity is calculated based on the mass of LIC7.2. **d** Cycling stability of the LIC7.2 electrode at a current of 40 μA cm$^{-2}$. The specific capacity is calculated based on the mass of LIC7.2. **e** Nyquist plots of the LIC7.2 electrode-based battery at the pristine state and after 10, 50, and 100 cycles. **f** XRD patterns of the LIC7.2 electrode at pristine, discharged, recharged states and after 100 cycles.

that the redox reaction of LIC7.2 contributes to a specific capacity of 50 mAh g$^{-1}$ after 10 cycles in the $I_2$/LIC7.2 electrode.

Based on the above analysis, it can be inferred that the redox behavior of LIC7.2 comprises two distinct components: the In$^{3+}$/In$^{2+}$ redox and the Cl$^-$ redox. In the $I_2$/LIC7.2 composite, a consistent In$^{3+}$/In$^{2+}$ redox reaction is observed, while the Cl$^-$ oxidation of LIC7.2 is notably suppressed due to the strong interaction between I$^+$ and Cl$^-$ ions. This difference in behavior suggests that the activation of $I_2$/I$^+$ redox reaction alters the Cl$^-$ redox activity of LIC7.2. The reduction or oxidation of solid electrolytes is generally believed to compromise their conductivity, leading to poor lithium-ion transfer within batteries. Contrarily, the sole LIC7.2 electrode maintains stable cycling performance over 100 cycles, with no significant changes in battery resistance observed after extensive cycling, as shown in Fig. 4e. These observations prove that the redox behavior of LIC7.2 does not adversely affect its ionic conductivity. Previous studies have indicated that solid electrolytes, such as Li$_6$PS$_5$Cl, initially undergo lithiation/delithiation processes, subsequently converting into thermodynamically stable but poorly conductive decomposition products[27,28]. These products hinder ionic transport across the electrode/solid electrolyte interface. However, unlike sulfide-based electrolytes, the redox reaction in LIC7.2 does not lead to its direct decomposition, as evidenced by the XRD patterns of the LIC7.2 electrode at pristine, charged, discharged and 100-cycled states shown in Fig. 4f. Moreover, the LIC7.2 in the $I_2$/LIC7.2 composite electrode can also maintain good structural stability during cycling, as confirmed by XRD results (Supplementary Fig. 14). Note that a slight shift of the XRD peaks is observed for the charged and discharged samples, further proving a reversible lithiation/delithation behavior of the LIC7.2. This good stability is further supported by the EIS results for the LIC7.2 electrode-based battery during its first cycle, presented in Supplementary Fig. 15. The overall resistances in different states of the battery (pristine, discharged, and charged) show comparable values, indicating that LIC7.2 maintains rapid lithium-ion kinetics throughout the redox process.

In typical ASS batteries, catholytes primarily serve as pathways for rapid ion transport. When the cutoff voltage exceeds the thermodynamic stability window of the electrolyte, severe decomposition can occur, contributing to irreversible capacity and obstructing efficient ion transport at the electrode/solid electrolyte interface. In contrast, LIC7.2 not only facilitates efficient ionic diffusion but also contributes reversible capacity through its redox activity while activating the $I_2$/I$^+$ redox couple. This triple functionality enhances the performance of the composite $I_2$/LIC7.2 electrode, underscoring the strategic advantage of employing LIC7.2 as the catholyte in ASS Li‖$I_2$ batteries.

## Electrochemical performance of the four-electron ASS Li‖$I_2$ battery

To evaluate the electrochemical performance of the as-designed ASS Li‖$I_2$ batteries based on $I_2$/LIC7.2 electrode, a series of galvanostatic cycling tests were conducted. Unless otherwise specified, all capacity values below are calculated based on the mass of $I_2$. Supplementary Fig. 16 displays the voltage profiles and long-term cycling stability of the battery at an $I_2$ mass loading of 0.5 mg cm$^{-2}$ at RT. The battery delivers a high capacity of 430 mAh g$^{-1}$ for the second cycle, and still maintains a capacity of ~400 mAh g$^{-1}$ after 200 stable cycles, with a capacity retention of 93%. Because LIC7.2 contributes to the cell capacity during cycling, the specific capacity based on the total mass of $I_2$ and LIC7.2 is also presented in the figures. It is worth noting that an average coulombic efficiency of 99.03% is achieved for the battery from 5th to 200th cycle, validating the advantages of ASS Li‖$I_2$ batteries over liquid-based Li‖$I_2$ batteries which usually show much lower coulombic efficiencies due to the severe polyiodides shuttle effect. This advantage is further demonstrated by the self-discharge performance. A very stable voltage of 3.2 V versus Li$^+$/Li could be maintained for 120 h after the battery assembly, which indicates there is no self-discharge (Supplementary Fig. 17a). When the battery is charged to 4 V

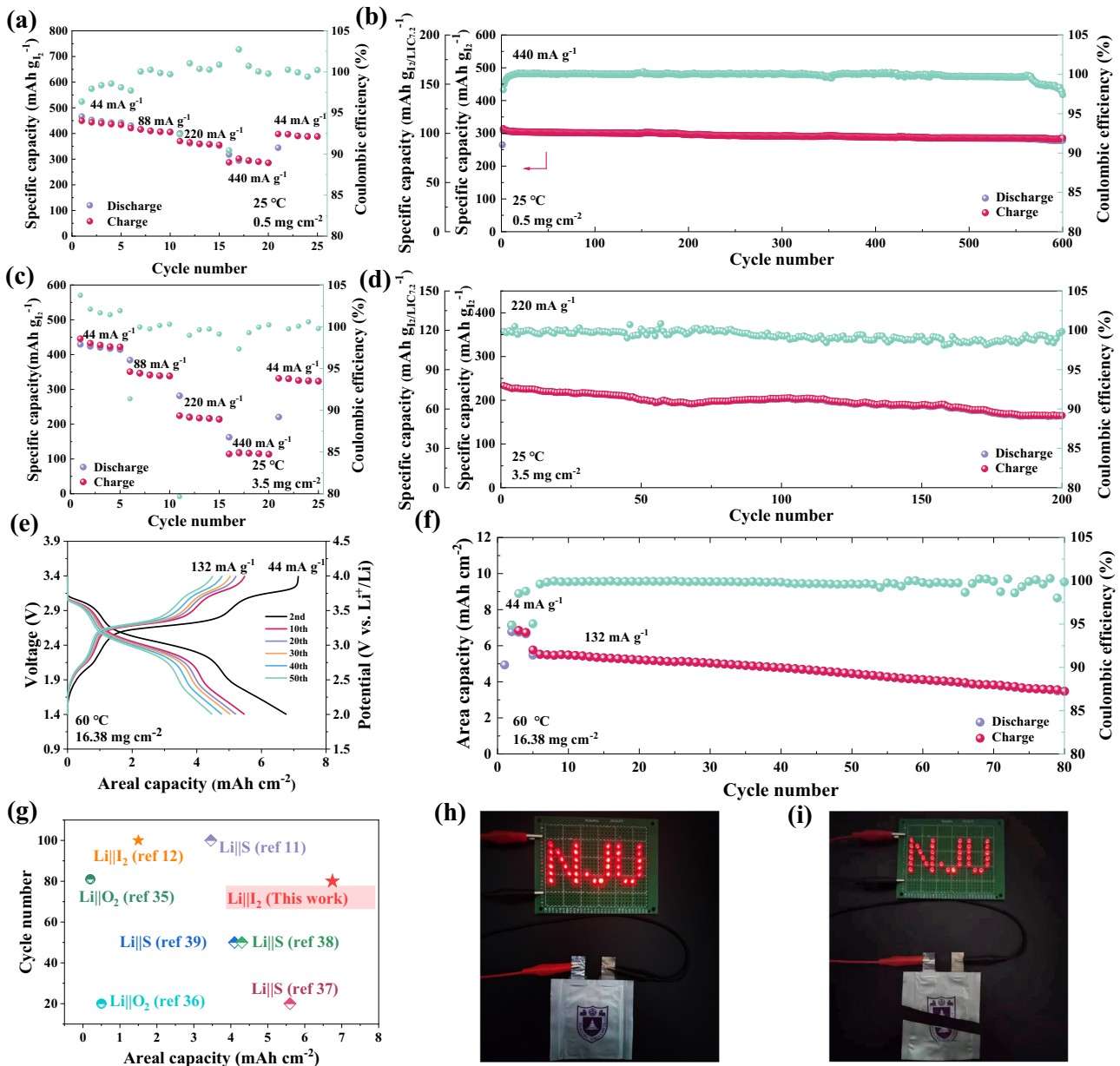

**Fig. 5 | Electrochemical performance of the four-electron ASS Li‖I₂ batteries.**
**a** Rate performance of the battery with a mass loading of 0.5 mg cm⁻² at RT.
**b** Cycling stability with the corresponding coulombic efficiency of the battery at 440 mA g⁻¹ and RT. The I₂ mass loading is 0.5 mg cm⁻². **c** Rate performance of the battery with a mass loading of 3.5 mg cm⁻² at RT. **d** Cycling stability with the corresponding coulombic efficiency of the battery at 220 mA g⁻¹ and RT. The I₂ mass loading is 3.5 mg cm⁻². Voltage profiles (**e**) and **f** cycling performance of the battery under high temperature (60 °C) and high mass loading (16.38 mg cm⁻²) at a current of 132 mA g⁻¹. **g** Comparison of areal capacity and cycle life of four-electron ASS Li‖ I₂ batteries and literature-reported results on ASS batteries with conversion-type electrodes under high areal capacity condition. **h** Photograph of the ASS Li‖I₂ pouch cell (40 × 40 mm film-type battery in a 60 × 60 mm case) powering LED lights. **i** Illustration of the pouch cell showing high safety and well-running under the condition of being half-cut.

versus Li⁺/Li and then rests for 120 h, the voltage slowly drops to 3.75 V versus Li⁺/Li and then keeps stable (Supplementary Fig. 17b). The resulting coulombic efficiency of 93.03% surpasses that of reported I₂ electrodes in liquid Zn‖I₂ batteries[29–31]. The ASS Li‖I₂ battery also displays exceptional rate performance (Fig. 5a), obtaining high discharge capacities of 449 mAh g⁻¹, 420 mAh g⁻¹, 370 mAh g⁻¹, and 302 mAh g⁻¹ at 44 mA g⁻¹, 88 mA g⁻¹, 220 mA g⁻¹, and 440 mA g⁻¹ respectively. When the rate is turned back to 88 mA g⁻¹, it still shows a discharge capacity of 400 mAh g⁻¹. To evaluate the fast and stable kinetics of I⁻/I₂/I⁺ chemistry, the battery is cycled at a high rate of 440 mA g⁻¹ and delivers a high capacity of 278 mAh g⁻¹ with a satisfying capacity retention of 91% for over 600 cycles (Fig. 5b). When the ASS Li‖I₂ battery is cycled at a voltage range of 2.4–4 V versus Li⁺/Li to exclude the capacity

contribution of LIC7.2 catholyte, it shows a lower capacity of 350 mAh g⁻¹ for the second cycle and maintains a capacity of ~200 mAh g⁻¹ after 400 cycles at 88 mA g⁻¹ and RT (Supplementary Fig. 18). This means that partial capacity of the I⁻/I₂ redox couple is also sacrificed at this narrower operation voltage window. Based on the above high performance, it is evident that cycling the battery at a voltage window of 2–4 V versus Li⁺/Li is essential to utilize the full capacity of four-electron I⁻/I₂/I⁺ chemistry, and the redox behavior of LIC7.2 does not affect the stability of the battery and efficient ionic transport at all. To further demonstrate the good compatibility of the LIC7.2 catholyte with I₂ electrode, galvanostatic cycling was also conducted for I₂/LYB electrode-based ASS Li‖I₂ battery (Supplementary Fig. 19). A lower capacity of ~250 mAh g⁻¹ is observed, corresponding to

the two-electron $I^-/I_2$ process, which however decays to 170 mAh g$^{-1}$ after only 8 cycles.

The electrochemical performance of the $I_2$/LIC7.2 based ASS Li‖$I_2$ battery was further evaluated at harsh conditions such as high areal capacities and high temperatures. At an $I_2$ mass loading of 3.5 mg cm$^{-2}$, a high discharge capacity of ~400 mAh g$^{-1}$ is obtained for the battery operating at 22 mA g$^{-1}$ and at RT (Supplementary Fig. 20), corresponding to a high areal capacity of 1.42 mAh cm$^{-2}$. A good rate performance is also demonstrated at this high mass loading condition and at RT (Fig. 5c). Noting that the rate capability at 3.5 mg cm$^{-2}$ is decreased compared to that at a lower mass loading of 0.5 mg cm$^{-2}$. This is attributed to the intrinsically low Li$^+$ diffusion coefficient of the conversion-type $I_2$ electrode. Moreover, at a high rate of 220 mA g$^{-1}$, it shows a satisfactory cycling stability and 83.6% capacity retention after 200 cycles (Fig. 5d). EIS measurements were conducted to monitor the evolution of battery resistance over this long-term cycling (Supplementary Fig. 21a). The electrolyte resistance (R1) is 50 Ω for the fresh battery, and keeps almost unchanged over 100 cycles and slightly grows to 73 Ω after 200 cycles. The semicircle at middle-frequency range is difficult to distinguish for the fresh battery, indicating a very intimate $I_2$/LIC7.2 electrode/electrolyte interface contact. After 50 cycles, a semicircle could be observed in the EIS curve of the battery, corresponding to an interface resistance (R2) of ~10 Ω, which gently increases to ~80 Ω after 200 cycles. The corresponding DRT analysis is displayed in Supplementary Fig. 21b. No obvious change in the peak intensity can be found at the short time constant range ($10^{-7}$–$10^{-2}$ s). An increasement of the intensities is observed for peaks at time constants above $10^{-2}$ s after 200 cycles, while the peak representing the ionic diffusion within $I_2$/LIC7.2 electrode with the longest time constant of ~1 s shows the biggest increasement. The most plausible reason for the increased interface resistance would be the repeated volume change of the electrode during conversion reactions, which could lead to the worsening of ionic transport within the $I_2$/LIC7.2 electrode and gradual physical deterioration of the electrode/electrolyte interface[32,33]. This also explains the relatively worse capacity retention of the battery with a higher mass loading (16.38 mg cm$^{-2}$, Fig. 5e, f) which experiences a more severe volume change than that with a low mass loading (0.5 mg cm$^{-2}$, Fig. 5b). At a harsh condition of a mass loading of 16.38 mg cm$^{-2}$ and an elevated temperature of 60 °C (Fig. 5e, f), an areal capacity of 6.75 mAh cm$^{-2}$ (412 mAh g$^{-1}$ based on $I_2$ and 137.4 mAh g$^{-1}$ based on $I_2$/LIC7.2) is achieved at a current of 44 mA g$^{-1}$. Even at this high mass loading and high-temperature test condition, this battery shows a satisfying cycling stability over 80 cycles with a capacity retention of 72.2%. It should be noted that bare $I_2$ easily sublimates at temperature higher than 40 °C. The effective trapping of $I_2$ inside the pores of KB avoids $I_2$ loss at 60 °C in the $I_2$/LIC7.2 composite electrode (Supplementary Fig. 22), and leads to the above good high-temperature performance.

Compared to conventional intercalation-positive electrodes, conversion-type positive electrodes such as $O_2$, S, and $I_2$ usually possess much higher specific capacities due to their ability to manifest multiple-electron chemistry. While S and $O_2$ have super high capacities (1672 mAh g$^{-1}$ for S based on S mass and over 5000 mAh g$^{-1}$ for $O_2$ based on the host material mass)[34], their application in ASS batteries still confronts severe challenges such as sluggish kinetics and huge overpotential. Thanks to the fast and stable four-electron $I^-/I_2/I^+$ conversion chemistry, the ASS Li‖$I_2$ batteries exhibit much better kinetics and cycling stability than ASS Li‖S and Li‖$O_2$ batteries, especially under a high areal capacity condition. In Fig. 5g our designed ASS Li‖$I_2$ batteries and literature-reported ASS Li‖S/$O_2$ batteries under high areal capacity condition are plotted in two dimensions of areal capacity and cycle number. Compared to two-electron conversion ASS Li‖$I_2$ battery based on poorly conductive hybrid electrolyte[12], the four-electron ASS Li‖$I_2$ battery based on LIC7.2 catholyte is able to operate at a six times

higher areal capacity (6.75 mAh cm$^{-2}$). Despite the high theoretical discharge capacity of $O_2$ electrode, current ASS Li‖$O_2$ batteries[35,36] are only able to operate at areal capacities less than 0.5 mAh cm$^{-2}$. Even compared with intensively studied ASS Li‖S batteries[11,37–39] our ASS Li‖$I_2$ battery still performs better at extreme areal capacity conditions. Last but not least, we further fabricated a pouch cell with dimensions of 40 × 40 mm and a capacity of 22 mAh to validate the potential application prospect of our ASS Li‖$I_2$ battery (Supplementary Fig. 23). It is able to power LED lights (Fig. 5h) and still shows well-running after being half-cut (Fig. 5i). While the ASS Li‖$I_2$ battery demonstrates high electrochemical performance and high safety, it is worth noting that the active $I_2$ mass in the $I_2$/LIC7.2 electrode only accounted for ~25%. Further increasing the $I_2$ mass ratio to 40% leads to a very limited activation of the $I_2$/I$^+$ redox (Supplementary Fig. 24), which is attributed to the insufficient I–Cl coordination environment near the $I_2$ particles, as well as a worse ionic conductivity. To further improve the practical capacity of the $I_2$ positive electrode, it's crucial to develop advanced $I_2$ composite electrode structures with higher specific surface area and improved electronic/ionic conductivity. Simultaneously, investigating the I–Cl coordination mechanism in the solid-state system would aid in understanding the interplay between $I_2$ and the catholyte, which helps to optimize the usage of active $I_2$ and catholyte in the electrode and achieve higher energy density for ASS Li‖$I_2$ batteries.

In summary, we have achieved an $I^-/I_2/I^+$ four-electron solid-phase conversion chemistry for ASS Li‖$I_2$ battery through the employment of chlorine-rich catholyte LIC7.2. The as-synthesized LIC7.2 with high local Cl$^-$ concentration provides sufficient coordination environment for the interhalogen interaction between I$^+$ species and Cl$^-$, effectively activating the complete $I_2$/I$^+$ conversion. Our proposed mechanism for this $I_2$/I$^+$ conversion involves an interface-mediated "heterogeneous oxidation" process, wherein the oxidation process occurs on the $I_2$ phase and the delithiation process occurs on the LIC7.2 catholyte phase. In contrast to typical catholytes, which primarily function as ionic pathways, the LIC7.2 catholyte showcases an additional capability —it actively contributes to the overall capacity through its redox activity, without compromising its efficient ionic conduction properties. Moreover, the utilization of highly conductive inorganic catholyte and electrolyte facilitates a direct solid-phase conversion between $I^-$, $I_2$, and $I^+$ at RT. Consequently, ASS Li‖$I_2$ batteries effectively circumvent the polyiodide shuttle effect commonly observed in liquid Li‖$I_2$ batteries. This advancement results in a high discharge capacity of 449 mAh g$^{-1}$ at 44 mA g$^{-1}$ and long-term stability over 600 cycles at 440 mA g$^{-1}$ at RT. Furthermore, the battery demonstrates stable cycling under practical high areal capacity (6.75 mAh cm$^{-2}$) and elevated temperature (60 °C) conditions, while rigorous pouch cell cutting tests confirm its high safety profile. This study underscores the promising application potential of ASS Li‖$I_2$ batteries in energy storage, particularly in scenarios requiring both high capacity and enhanced safety features, and would definitely garner increased attention towards the research of ASS batteries based on novel conversion positive electrodes.

## Methods
### Material synthesis
Ball milling was used to prepare the $I_2$@KB powder. $I_2$(Aladdin) and KB (EC-600JD) with a weight ratio of 1:1 were ball milled at 500 rpm for 12 h. The LIC6, LIC6.6, LIC7.2, LIC7.8, and LYB were synthesized by a one-step ball-milling process. The LIC6 and LIC7.2 were synthesized by ball-milling stoichiometry LiCl (Sigma Aldrich, 99%) and InCl$_3$ (Alfa Aesar, 99.99%) at 600 rpm for 24 h. The LYB was synthesized by ball-milling stoichiometry LiBr (Sigma Aldrich, 99.9%) and YBr$_3$ (Alfa Aesar, 99.9%) at 600 rpm for 24 h. The LGPS was purchased from Shenzhen MTI. The crystalline LIC6 was prepared through a water-mediated method reported elsewhere[40]. To prepare the $I_2$/LIC6, $I_2$/LIC6.6, $I_2$/

LIC7.2, $I_2$/LIC7.8, and $I_2$/LYB composite electrodes, the $I_2$@KB powder with ~48 wt% $I_2$ and the catholytes was ball milled at 500 rpm for 12 h with a certain weight ratio (3:7 for 15 wt% $I_2$, 1:1 for 25 wt% $I_2$ and 4:1 for 40 wt% $I_2$). LIC6/LIC6.6/LIC7.2/LIC7.8/LYB-based electrodes were prepared by ball milling them with KB powder with a weight ratio of 1:2 at 400 rpm for 4 h. All the above ball-milling processes were performed using 45 mL WC jars containing 1 mm WC balls with a ball-to-powder ratio of 40:1. The hand-ground $I_2$/LIC7.2 electrode was prepared by milling the $I_2$@KB and LIC7.2 (weight ratio of 1:1) with a pestle and mortar for 30 min.

## Material characterizations

Raman spectroscopy was conducted using a confocal Raman instrument (NTEGRA Spectra AFM Raman Confocal SNOM). The chemical composition and surface states were analyzed through X-ray photoelectron spectroscopy (XPS) on a PHI 5000 VersaProbe-II system. EIS was performed with a Solartron 1287/1290 system, applying a 5 mV perturbation voltage in a frequency range of 1 MHz to 0.1 Hz. Measurements included six data points per decade, with a 15-min resting period for the cell prior to each test. The obtained EIS data were analyzed using ZPlot software, and the DRT was evaluated with EISART software[41]. Thermal stability was assessed through thermogravimetric analysis using a TA SDTQ600 analyzer. The temperature was increased from RT (25 °C) to 600 °C at a rate of 10 °C/min under a nitrogen atmosphere. Structural characterization was carried out via XRD on a PHI 5000 VersaProbe system. The morphological features of the electrodes and electrolyte were observed with a Hitachi SU8010 scanning electron microscope, while detailed microstructural analysis was performed using a FEI TF20 transmission electron microscope at an operating voltage of 200 kV.

## Cell assembly and electrochemical tests

Cell assembly was performed under inert conditions using an argon glove box, where the ASS batteries were constructed inside cylindrical housings made of PEEK material featuring a 10 mm inner diameter. First, 50 mg crystalline LIC6 with ionic conductivity over 1 mS cm⁻¹ was added to the mold and mechanically pressed at 300 MPa for 5 min. Then 50 mg LGPS powder as protective layer was spread on LIC6 and then pressed at 300 MPa for 5 min. $I_2$/LIC6, $I_2$/LIC6.6, $I_2$/LIC7.2, $I_2$/LIC7.8, and $I_2$/LYB composite electrode was distributed evenly on the side of the LIC6 pellet. The thickness of the 50 mg solid electrolyte layer is 243 μm (Supplementary Fig. 25) with a porosity of 3.71% (Supplementary Fig. 26). The thickness of the $I_2$/LIC7.2 with a typical mass loading of 5.1 mg cm⁻² is 59 μm with a porosity of 7.44% (Supplementary Fig. 27). An In foil (with a diameter of 8 mm, thickness of 100 μm) was attached to the other side of LGPS pellet. A Li foil with a molar ratio of Li:In = 0.5:1 was pressed on the In foil and kept under 50 MPa for 4 h (Lithium foil was purchased from China Energy Lithium Co, ≥99.9%,). The fabricated cells were rested for 6 h before testing. All cell assembly processes were carried out in an argon-filled glove box and cycled at a stacking pressure of 50 MPa. To manufacture solid-state pouch cells, LGPS, LIC6, and electrode films are prepared by mixing with PTFE (Daikin Industries) additives (mass ratio: ≈1%) and rolling into thin sheets (thickness: ≈0.05 mm). Pouch Li–In|LGPS-LIC6|$I_2$/LIC7.2 cells are made by pressing the LGPS, LIC6, and electrode films at 100 MPa and then pressing the Li–In foil onto one side of the LGPS film. The size of the LGPS and LIC6 films was 40×40 mm and the size of both the Li–In foil and the electrode film ($I_2$ loading: 62 mg) was 38×38 mm. Regarding the preparation of the positive electrode film, IKB, LIC7.2, and PTFE were firstly mixed in a mortar and pestle in a mass ratio of 50:50:1 to form the film. The initial film was manually rolled and cut out to 40 × 40 mm, then the film was continued to be thinned and enlarged by a roller press and cut to 40 × 40 mm, and

the rolling process was repeated about 3–4 times (the last time the electrode film was cut to 38 × 38 mm size). The pouch cells were then encapsulated in vacuum in an aluminum plastic film with a diameter of 60 mm. Galvanostatic charge/discharge tests of the ASS Li||$I_2$ battery were carried out by the Neware battery test system (CT-4008T, Shenzhen, China). The CV measurements were performed on the Bio-Logic VSP-300. Unless otherwise noted, all electrochemical tests were performed in a thermostatic test chamber maintained at a temperature of ~25 °C.

## DFT calculations

All computations used VASP package with PAW pseudo-potentials. Exchange-correlation energetics of electrons were computed utilizing PBE functionals within the GGA framework. The simulations maintained a consistent plane wave basis cutoff energy of 520 eV throughout all calculations. For all calculations in this work, the cutoff energy of the plane wave was set to be 520 eV. The convergence criteria for energy and force are set to be less than $10^{-5}$ eV and 0.02 eV Å⁻¹. Spin polarization is turned on for all calculations. LIC7.2 and LYB have the same crystal structure (space group: C2/m), and the (001) surface was considered to be the main exposed surface for catalysis in this work. The slab model was constructed using $2 \times 2$ supercells and a vacuum layer of 20 Å was applied in the z-direction of the slab models to avoid the effect of periodic boundaries. For the k-points sampling of Brillouin-zone integrals, the Monkhorst-Pack grids were set to be $2 \times 2 \times 1$. The cohesive energies were defined as

$$\Delta E = \frac{E_{sys} - E_{sub} - nE_{atom}}{n}$$

where $E_{sys}$, $E_{sub}$ and $E_{atom}$ represent the energies of the system, LIC7.2 or LYB substrate and isolated atoms such as Li, I, Cl, Br; and $n$ means the number of isolated atoms. The selection criterion of the two-dimensional charge density sections is to display the charge information of I–Cl and I–Br as much as possible. Thus, all the two-dimensional charge density sections were cut from the center of I and its most adjacent Cl or Br atoms. All charge density maps were drawn with the same parameters. The isosurface level was set to 0.08 e/bohr³. For more accurate charge information of I–Cl and I–Br, the Bader charge was calculated. The employed DFT structures can be seen in Supplementary Data 1–6.

## Data availability

The datasets generated and analyzed in this work are included in this article and Supplementary Information. Source data are provided with this paper.

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

## Acknowledgements

This research was supported by the National Key R&D Program of China (2021YFB3800300—P.H. and H.Z.), the National Natural Science Foundation of China (923722201—H.Z., 22179059—P.H., 22239002—H.Z.), Key R&D project funded by department of science and technology of Jiangsu Province (BE2020003—P.H.), science and technology innovation fund for emission peak and carbon neutrality of Jiangsu province (BK20231512—P.H., BK20220034—H.Z.).

## Author contributions

P.H. and H.Z. conceived the idea and supervised the research. Z.C. and H.L. conducted the experiments. M.Z. performed the DFT calculations. Z.C., H.L., and P. H. analyzed the experiment results and wrote the manuscript. H.P., C.S., W.L., and M.W. discussed the results and commented on the manuscript. Z.C. and H.L. contributed equally to this work.

## Competing interests

The authors declare no competing interests.
