## [Peer Review File · Nature Communications]

Realizing four-electron conversion for all-solid-state Li-I₂ batteries at room temperature

Corresponding Author: Professor Haoshen Zhou

Version 0:

Reviewer comments:

Reviewer #1

(Remarks to the Author)

In this work, the authors synthesized Cl-rich catholyte to pair with I₂ cathode for activating the I₂/I⁺ redox couple in solid-state battery, which delivers better performance compared with liquid-electrolyte analogues. Despite the exceptional electrochemical data and substantial amount of characterizations applied, the reviewer found the underlying mechanism for the activation in I₂/LIC417 composite cathode is still vague. Specifically, a clearer view of how the structure stabilizes in I₂ and LIC417 during charging when the oxidation and lithium extraction occur in different kinds of particles is needed. Below are some comments for the authors to consider.

Comments:

1. What was the stoichiometry of Li_{4.2}InCl_{7.2} selected based on? Can the authors show how does the activity of I₂/I⁺ couple change with different Cl⁻ content?
2. The XRD in Figure S1 needs to be properly analyzed. Which reference is used to identify the structure? The reviewer suggests the authors perform Rietveld refinement on the XRD to compare the structural differences between LIC316 and LIC417.
3. In Figure 2b, since all the lithium source is contributed by the catholyte, the specific capacity should be based on the mass of the composite electrode instead of I₂ species only. How is specific capacity compared to the theoretical capacity of I₂/I⁺ redox couple when considering the weight of catholyte? How is the weight ratio between I₂ and catholyte affecting the performance?
4. What is the particle size of each component after balling? In Figure 2e, the model only explains how the interfaces between I₂ and LIC417 gets stabilized. But inside the I₂ particles or LIC417 particles, how are the I⁺ and Cl⁻ ions supposed to interact with each other? Is there ion interdiffusion happening across the interface?
5. Why are the Raman spectra at stage b and stage d in Figure 3b looking exactly the same? It is hard to believe even the noise looks the same as well, especially after the electrode has gone through discharging and charging.
6. In Figure 5b, how was the coulombic efficiency calculated? Why would there be capacity degradation to 93% at cycle #200 when the average CE is claimed to be 100%?

Reviewer #2

(Remarks to the Author)

This work reports the application of high-capacity four-electron solid-conversion I⁻/I₂/I⁺ chemistry in solid-state lithium metal batteries, utilizing a chlorine-rich Li_{4.2}InCl_{7.2} ionic conductive catholyte to facilitate I-Cl bond formation. The manuscript includes extensive characterization to elucidate the four-electron redox mechanism and conversion kinetics. However, some critical aspects, such as the phase evolution of Li_{4.2}InCl_{7.2} upon I-Cl interhalogen bond formation and the solid-solid conversion efficiency, were not adequately addressed. Therefore, I cannot support its publication in its current form.

1. Since Li_{4.2}InCl_{7.2} is a Li⁺ ion conductor with immobilized chloride ions in its crystal structure, the I₂/I⁺ conversion efficiency and I-Cl bond formation likely depend on the contact between I₂ and Cl. How can effective contact be maintained while retaining a relatively high iodine content in the electrode?

2. Determining the phase evolution of Li_{4.2}InCl_{7.2} during charge/discharge cycles is necessary to understand the I-Cl interhalogen bond formation process. If the Li_{4.2}InCl_{7.2} phase is maintained during redox, as claimed in the manuscript, the electronegativity of chloride might decrease due to ionic or covalent bonding in the solid structure, potentially reducing its capacity to stabilize I⁺ species.

3. Two distinct charge plateaus are observed for the initial charge of I₂/LIC417 or I₂/LIC316 electrodes (Figure 2b and Figure S4). Does this indicate that the valence state of iodine in the I₂/LIC417 or I₂/LIC316 compounds is not 0? Please provide a detailed explanation.
4. The manuscript mentions a Raman shift of I₂ at 167 cm⁻¹ observed in pristine I₂/LYB and I₂/LIC417 cathodes (Figure 2c). However, this corresponds to the characteristic peak of I₅⁻ in other literature (J. Am. Chem. Soc. 2024, 146, 10, 6744–6752; Adv. Mater. 2022, 34, 2201716). Why is this the case? Additionally, a peak around 100-130 cm⁻¹ in the Raman spectrum, which changes during the redox reaction (Figure 2c and Figure 3b), should not be ignored.
5. The self-discharge performance of the four-electron Zn-I₂ full battery at different depths of discharge (DOD) should be provided.
6. XRD patterns after ten or twenty cycles, or for LIC417 in different states in I₂/LIC417 cathodes, should be included.
7. In Figure 5c, the battery operates at 60 °C, exceeding the sublimation temperature of iodine. The impact of potential iodine phase changes on the battery's performance warrants discussion.
8. Why was LIC316 chosen as the solid-state electrolyte rather than LIC417? Does Cl⁻ migrate from LIC417 to LIC316 under the concentration gradient?

Reviewer #3

(Remarks to the Author)

- The manuscript prepared by Cheng et al. reports an all-solid-state Li-I₂ battery. The authors proposed that the battery operates based on I⁻/I₂⁺ (four-electron) redox. The authors reported detailed studies on the battery mechanism and cycling performance. Before the manuscript can be considered for publication, the following comments still need to be addressed:
1. The specific capacity reported in this manuscript was based on the mass of I₂. According to the author, the LIC417 plays an important role in helping the battery achieve a higher capacity by interhalogen interaction between Cl⁻ and I⁺. Therefore, the mass of LIC417 should also be included in the calculation of specific capacity, not just the I₂ itself. The author should also report energy density calculated in this manner, in addition to specific capacity and areal capacity.
 2. The cycling data reported in Figure 5 has I₂ loading of 0.5 mg cm⁻². This is too low, and it is suggested that the author should report the 3.5 mg cm⁻² data in the main figure instead.
 3. The rate capability of the battery when the loading is increased (3.5 mg cm⁻² vs. 0.5 mg cm⁻²) is dramatically decreased. The author should explain the reason for such decay in rate capability.
 4. Can the Cl content be further increased in LIC417? Will this further improve the battery performance? Or is LIC417 the optimal composition?
 5. The XRD data present in Fig. 4f actually shows differences. Compared to the Pristine data, the After 1st discharge and After 1st charge XRD data have their XRD peaks slightly shifted to higher angles. Also, the peak at ~ 72° in pristine XRD data seems to disappear after the battery has cycled. The author should explain these differences. In addition, does the XRD pattern of LIC417 further changes when the battery has cycled for a while (ex. cycle 100)?
 6. In line 253, the sentence should refer to Fig. 3c, not Fig. 3d. Same for line 237, there is a mistake in figure referencing.

Reviewer #4

(Remarks to the Author)

The authors demonstrate an all-solid-state Li-I₂ battery that enables four-electron conversion chemistry (I⁻/I₂⁺). Poor interfacial contact between porous cathode and solid-state electrolyte has generally been a significant issue, causing poor performance in all-solid-state batteries. In this work, the authors introduce a chlorine-rich lithium-ion conductor, LIC417 (Li₄.2InCl₇.2) as a catholyte. This addition to form I₂/LIC417 composite cathode not only enhances Li-ion conduction pathway through improved interfacial contact but also promotes a second major redox chemistry (I₂ ↔ 2I-Cl) in addition to the commonly known 2I- ↔ I₂ conversion. This strategy contributes to enhanced capacity (almost twice as high as that of a conventional lithium-iodide battery) and improved cycle life at room temperature. To the best of our knowledge, this is the first demonstration of all-solid-state battery based on 4 electron interhalogen battery chemistries. The claims and conclusions made by the authors are all well supported by adequate datasets with thorough analysis and interpretation. We recommend this paper be published in Nature Communication after some minor revisions.

1. In lines 146~148 of the main text, the authors states, "In the LIC417-based system, LIC417 is used as the catholyte, whereas the bulk solid-state electrolyte (SSE) remains LIC316 to ensure fast ionic transport within the battery". Figure 1 also depicts the battery cell structure, supporting this statement. However, a few pages later, the authors unexpectedly interpret their DRT data with the interfaces generated by LGPS (lines 273~275). They briefly mention that LIC316 and LGPS are used as a dual-layer electrolyte here, which is inconsistent with the earlier cell description and figure drawing presented in the manuscript. This could potentially confuse readers. Please revise cell-drawing in Figure 1 and the main text (lines 146~148) to accurately describe the Li-I₂ battery cell studied in this paper. Please also include plausible reasons for choosing LIC316-LGPS bilayer over a single layer of LIC316 or LGPS (What is the main role of LGPS in this system? How would the cell behave if it was assembled with a single layered LIC316 SSE or LGPS SSE, in combination with I₂/LIC416 composite cathode?)
2. Please provide the thickness of the electrodes and solid-state electrolytes along with accurate cell stack/voice space information for those who might want to reproduce the data.

3. The authors commented in lines 257-259 that the active I₂ mass in the I₂/LIC417 cathode is only about 25%, indicating room for improvement. To achieve higher I₂ loading beyond 25 wt%, it is crucial to supply sufficient chlorine to form the interhalogen charge product (I-Cl). However, increasing the iodine mass loading would reduce the relative mass loading of other cathode components including LIC417 catholyte, a source of chlorine. Therefore, there might be a theoretical limit to the possible proportion of I₂ in the cathode that enables the complete 4 electron conversion from the system. It would be very helpful if the authors could provide additional insights into this point based on their scientific understanding of the I-Cl forming mechanism and the interaction between LIC417 and I₂ (stoichiometric ratio?) in the main text. If the authors already have data showing the effect of increasing the proportion of iodine in the cathode (from 25 wt% to 30 ~ 50 wt%), please include it in the supporting information. This would provide valuable information and guidance for further research in all-solid-state lithium-iodide battery development.

4. There are several typos and mislabelling errors in the figures

- In Figure 2 caption: (f) should be changed to (g) for Two dimensional charge density distribution.
- In line 198, please add (Figure 2f) at the end of the sentence ending with LYB.
- In line 199, Figure 2f should be changed to Figure 2g.
- In line 237, delete (Figure 3c)
- In line 253, Figure 3d should be changed to Figure 3c.
- In line 264, please add the full name of DRT
- In line 267, Figure 3e and 3f should be changed to Figure 3d and 3e
- In line 288, Figure 3g should be changed to Figure 3f

5. In line 422, please include the specific capacity value next to the areal capacity of 6.75 mAh cm⁻² for easier comparison with other systems.

Reviewer #5

(Remarks to the Author)

"I co-reviewed this manuscript with one of the reviewers who provided the listed reports. This is part of the Nature Communications initiative to facilitate training in peer review and to provide appropriate recognition for Early Career Researchers who co-review manuscripts."

Version 1:

Reviewer comments:

Reviewer #1

(Remarks to the Author)

The authors have made major improvements on the manuscript. Most of my comments were addressed. There are a few more questions below to answer before this manuscript can be published.

1. The authors need to make the statement clear regarding the specific capacity in the beginning of the manuscript. "... Based on the I⁻/I₂/I⁺ four-electron chemistry, the as-designed all-solid-state Li-I₂ batteries deliver a high specific capacity of 449 mAh g⁻¹ based on I₂ mass and an impressive cycling stability over 600 cycles with a capacity retention of 93% at 1C and at room temperature..." To avoid confusion, the authors should add the rate at which the high specific capacity of 449 mAh g⁻¹ was achieved.
2. The specific capacity based on the mass of I₂ and LIC should be labeled on Fig 2b for a fair comparison with Fig 2a.
3. The activation mechanism for I₂/I⁺ is still vague at this point. The mechanism of Co/Li₂O system does not really apply in this case. Co atoms bond strongly with O atoms after delithiation, but how did iodine atom bond with LIC? Is it covalent bond or ionic bond?
4. In Fig 5d, the I₂/LIC7.2 cathode is only able to deliver a specific capacity of 80 mAh/g with I₂ mass loading of 3.5 mg cm⁻² at 0.5C, if you consider the mass of I₂ and LIC7.2. How is authors' view on the practical use of such cathode in ASSB?

Reviewer #2

(Remarks to the Author)

The authors have emphasized the charge mechanism in the revised manuscript, stating that "during the charging process, the lithium halide electrolyte (LIC7.2) releases lithium ions, and the halide ions bind to the iodine on the surface of the iodine particles. At the same time, the iodine molecules lose electrons and form interhalogen bonds." However, it is clear that chloride ions remain ionically bonded in the delithiated catholyte phase, as confirmed by the XRD pattern (Fig.R9). Previous studies on the formation of the ICl interhalogen phase in battery systems have shown that this process depends on the electronegativity of chloride ions, with free chloride ions being more favorable for this formation. Therefore, it is crucial for the authors to provide further evidence demonstrating that I⁺ binds with the surrounding delithiated LIC7.2 to form interhalogen bonds. This is a critical point that directly impacts the validity of the findings presented in this work.

The supplemented Fig. R12 in the response letter showed a rapid OCV decay after 120 hours. Although this was observed in a solid-state cell where both electrodes are solidified, the underlying reasons for this decay must be addressed. It is crucial to investigate potential causes, including solid-solid interface stability issues, ion transport limitations, or degradation

of active materials, to offer a comprehensive understanding of this phenomenon.

Reviewer #3

(Remarks to the Author)

The authors have addressed my previous comments well and revised their manuscript accordingly. The manuscript looks good now and is suggested to be accepted for publication.

Reviewer #4

(Remarks to the Author)

The authors have thoroughly addressed all the concerns and comments raised by the reviewers, and I recommend this paper for publication in Nature Comm.

Reviewer #5

(Remarks to the Author)

Version 2:

Reviewer comments:

Reviewer #1

(Remarks to the Author)

The authors have addressed my comments. The manuscript can now be published.

Reviewer #2

(Remarks to the Author)

All of my comments are addressed now. I recommend this paper for publication in Nat. Commun.

Response to Reviewers' Comments

We would like to thank all the reviewers for their constructive comments and suggestions, which has in our view significantly raised the quality of the manuscript (NCOMMS-24-31566-T). We have modified the manuscript accordingly, and listed the detailed corrections below point by point for each reviewer. Moreover, all revised portion has been marked in yellow in the revised manuscript. The main corrections and the responses to the reviewers' comments are as follows.

Point-to-Point Responses (see next page)

Reviewer #1 (Remarks to the Author):

In this work, the authors synthesized Cl-rich catholyte to pair with I₂ cathode for activating the I₂/I⁺ redox couple in solid-state battery, which delivers better performance compared with liquid-electrolyte analogues. Despite the exceptional electrochemical data and substantial number of characterizations applied, the reviewer found the underlying mechanism for the activation in I₂/LIC417 composite cathode is still vague. Specifically, a clearer view of how the structure stabilizes in I₂ and LIC417 during charging when the oxidation and lithium extraction occur in different kinds of particles is needed. Below are some comments for the authors to consider.

Response: We appreciate the reviewer for this valuable and constructive comment. Please see the point-to-point responses below.

Comments:

1. What was the stoichiometry of Li_{4.2}InCl_{7.2} selected based on? Can the authors show how does the activity of I₂/I⁺ couple change with different Cl⁻ content?

Response: We thank the reviewer for raising this comment. **To obtain the optimized electrochemical performance of the I₂/I⁺ redox couple, both Cl⁻ concentration (to stabilize I⁺) and ionic conductivity (to provide fast ionic transport) of the LIC catholyte need to be taken into consideration.** We prepared the LIC by ball milling process with four different stoichiometry: Li₃InCl₆ (LIC6), Li_{3.6}InCl_{6.6} (LIC6.6), Li_{4.2}InCl_{7.2} (LIC7.2), and Li_{4.8}InCl_{7.8} (LIC7.8). Their XRD patterns and EIS results are shown in **Fig. R1a** and **R1b**. All of them can be indexed with a C2/m space group, while some minor peaks corresponding to LiCl shows in the XRD pattern of LIC7.8. This implies that the LIC7.8 stoichiometry is out of the Li-In-Cl solid solution range. With more Cl content, the ionic conductivity drops from 0.63 mS/cm for LIC6 to 0.06 mS/cm for LIC7.8.

We then prepared the I₂ composite cathode mixed with these different LIC catholytes and assemble them in all-solid-state (ASS) Li-I₂ cells, to evaluate their ability to activate I₂/I⁺ redox couple. As shown in **Fig. R1c**, the charging capacities for I₂/LIC6, I₂/LIC6.6, I₂/LIC7.2 and I₂/LIC7.8 cathode are 120, 186, 261 and 267 mAh g⁻¹, respectively. Therefore, an enough high Cl content in LIC (≥7.2) is important to fully activate the I₂/I⁺ redox reaction. To achieve both high ionic transport and full utilization of I₂/I⁺ redox couple, the LIC7.2 is selected as the optimized catholyte in this system. We have added the related discussion in line 136-158 on page 5-6.

Fig.R1 (a) XRD patterns and (b) Nyquist plots of the LIC6, LIC6.6, LIC7.2 and LIC7.8 at room temperature. (c) Voltage profiles of ASS Li-I₂ batteries based on different catholytes during the first charging at 0.05 C and at room temperature.

2. The XRD in Figure S1 needs to be properly analyzed. Which reference is used to identify the structure? The reviewer suggests the authors perform Rietveld refinement on the XRD to compare the structural differences between LIC316 and LIC417.

Response: As the reviewer suggested, Rietveld refinements are conducted for both LIC6 and LIC7.2 based on the structure information given by a previous study on Li₃InCl₆ from Xueliang Sun's group (*Angew. Chem. Int. Ed.* 2019, 58, 1 – 7). Both samples can be well refined with a same structure which belongs to the C2/m space group (**Fig. R2**). As seen by the refinement result table (Table. R1 and R2), these two samples have different lattice parameters. We have added the data in the supplementary information.

Fig.R2 Rietveld refinements of (a) LIC6 and (b) LIC7.2.

Table.R1 Rietveld analysis results for the XRD pattern of Li₃InCl₆.

$a = 6.4204(7) \text{ \AA}$ $b = 11.0746(5) \text{ \AA}$ $c = 6.3936(1) \text{ \AA}$ $V = 428.22(6) \text{ \AA}^3$ Space group C2/m						
Site	x	Y	Z	Fractional Occupancy	Uiso	Wyckoff site
Cl1	0.27175	0.16829	0.25479	1	0.01	8j
Cl2	0.77727	0.00000	0.27863	1	0.01	4i
In1	0.00000	0.16732	0.50000	0.53	0.01	4h
Li1	0.00000	0.27492	0.00000	1	0.01	4g
Li2	0.00000	0.00000	0.00000	1	0.01	2a

Table.R2 Rietveld analysis results for the XRD pattern of $\text{Li}_{4.2}\text{InCl}_{7.2}$.

$a = 6.4022(0) \text{ \AA}$ $b = 11.0627(8) \text{ \AA}$ $c = 6.3747(6) \text{ \AA}$ $V = 424.56(7) \text{ \AA}^3$ Space group C2/m						
Site	x	Y	Z	Fractional Occupancy	Uiso	Wyckoff site
Cl1	0.24216	0.16333	0.25462	0.875	0.01	8j
Cl2	0.72525	0.00000	0.26660	0.875	0.01	4i
In1	0.00000	0.17641	0.50000	0.375	0.01	4h
Li1	0.00000	0.27841	0.00000	1	0.01	4g
Li2	0.00000	0.00000	0.00000	1	0.01	2a

3. In Figure 2b, since all the lithium source is contributed by the catholyte, the specific capacity should be based on the mass of the composite electrode instead of I_2 species only. How is specific capacity compared to the theoretical capacity of I_2/I^+ redox couple when considering the weight of catholyte? How is the weight ratio between I_2 and catholyte affecting the performance?

Response: We agree with the reviewer that during the I_2/I^+ redox reaction, all the lithium source comes from the catholyte. After taking consideration of the catholyte mass (weight ratio between I_2 and LIC7.2 is 1:2), the specific capacity of the $\text{I}_2/\text{LIC7.2}$ cathode changes from 265 mAh g^{-1} to 88.3 mAh g^{-1} for the first charging process. We also studied the performance of $\text{I}_2/\text{LIC7.2}$ cathode during first charging with different I_2 mass loading ($\text{I}_2:\text{KB}:\text{LIC7.2}=15:15:70$, 15 wt% I_2 ; $\text{I}_2:\text{LIC7.2}=25:25:50$, 25 wt% I_2 ; $\text{I}_2:\text{LIC7.2}=40:40:20$, 40 wt% I_2). As shown in **Fig. R3**, under a I_2 mass loading of 15 wt% and 25 wt%, a full utilization of I_2/I^+ redox capacity can be obtained. This is due to enough Cl coordination environment around every I_2 particles at the presence of high content LIC7.2 in the $\text{I}_2/\text{LIC7.2}$ cathode. When the I_2 mass loading increases to 40 wt%, the charging capacity decreases to ~ 20 mAh g^{-1} . This is attributed to the insufficient catholyte amount in the $\text{I}_2/\text{LIC7.2}$ cathode, which leads to both insufficient I-Cl coordination environment and poor ionic transport. We added this discussion in the manuscript in line 507-510 on Page 20.

We would also like to point out that, although the lithium source is contributed by the LIC7.2 catholyte during I_2/I^+ redox, we didn't intentionally increase the amount of catholyte in the $\text{I}_2/\text{LIC7.2}$ cathode to

obtain four-electron conversion. The conversion-type cathode materials inherently require a relatively large amount of electrolyte, due to their poor ionic conductivities. For example, in the composite cathode of all-solid-state Li-S batteries, a typical usage of the catholyte and sulfur is 40-50 wt% and 20-30 wt%, respectively¹⁻⁴. In most systems, catholytes remain electrochemically inactive during cycling. However, the LIC7.2 catholyte used in the Li-I₂ system actively participates in the electrochemical process alongside I₂, while still maintaining its high ionic conductivity. We have simply maximized the capacity extraction based on this conventional amount of electrolyte, in order to enhance the capacity of the conversion reaction in all-solid-state electrodes.

Moreover, at a practical level, we would like to say that, improving the mass loading of the conversion-type cathodes is very important to achieve high energy densities. Especially, for four-electron I₂ cathodes, since the lithium source comes from the catholyte, the balance between practical capacity and usage of I₂/I⁺ redox need to be taken into consideration when designing future high-energy ASS Li-I₂ batteries.

Lastly, for clarification, we added the specific capacity calculated based on the mass of I₂/LIC7.2 to the figures in the revised manuscript.

Fig.R3 Voltage profiles of I₂/LIC7.2-based ASS Li-I₂ batteries with different I₂ mass loading.

1. Pan, H. et al. Carbon-free and binder-free Li-Al alloy anode enabling an all-solid-state Li-S battery with high energy and stability. *Sci. Adv.* **8**, eabn4372 (2022).
2. Hou, L.-P. et al. Improved interfacial electronic contacts powering high sulfur utilization in all-solid-state lithium-sulfur batteries. *Energy Storage Mater.* **25**, 436-442 (2020).
3. Liu, M. et al. Quantification of the Li-ion diffusion over an interface coating in all-solid-state batteries via NMR measurements. *Nat. Commun.* **12**, 5943 (2021).
4. Kim, J. T. et al. Manipulating Li₂S₂/Li₂S mixed discharge products of all-solid-state lithium sulfur batteries for improved cycle life. *Nat. Commun.* **14**, 6404 (2023).

4. What is the particle size of each component after balling? In Figure 2e, the model only explains how the interfaces between I₂ and LIC417 gets stabilized. But inside the I₂ particles or LIC417 particles, how are the I⁺ and Cl⁻ ions supposed to interact with each other? Is there ion interdiffusion happening across the interface?

Response: We thank the reviewer for raising this constructive comment.

Firstly, to investigate the size of each component, high-resolution TEM measurement is conducted on the ball-milled I₂/LIC7.2 mixture. As shown in **Fig. R4a**, the I₂ domain and LIC7.2 domain are of several nanometers. EDX element mapping also proves a homogenous mixing of I₂/LIC7.2 at the nanoscale (**Fig. R4b**). The extensive nanointerface between I₂ and LIC7.2 facilitates the release of the majority of the I₂/I⁺ redox capacity, enhancing the conversion efficiency of the I₂/I⁺ reaction. **This interface-mediated heterogeneous oxidation mechanism has been reported and widely acknowledged for conversion-type metal oxide electrode materials of Li-ion battery⁵⁻⁷.** For example, Yi Cui et al. investigated the oxidation mechanism of Co/Li₂O composite⁶. They proved that when the material's size is at the micron scale, its charging capacity is quite limited (<100 mAh g⁻¹), and only at the nanoscale can it achieve a significantly large specific capacity during charging (over 600 mAh/g). During charging, nanoscale Co undergoes an **interface-mediated heterogeneous oxidation** induced by the interface with Li₂O. As Li₂O releases lithium ions, oxygen ions bind to the Co atoms on the surface of the metallic Co, with these Co atoms simultaneously losing electrons, forming Co₃O₄ (equation R1). Of course, both metallic Co and the charging product Co₃O₄ have poor ionic conductivity for oxygen ions. However, Co/Li₂O can still achieve a high charging capacity and a considerable amount of Co₃O₄ conversion. This is the result of interfacial-induced heterogeneous oxidation reactions at the nanoscale.

A similar reaction occurs in our study as shown in equation R2. Nanoscale iodine and lithium halide electrolyte also form a pair of charging reactants. During the charging process, the lithium halide electrolyte (LIC7.2) releases lithium ions, and the halide ions bind to the iodine on the surface of the iodine particles (**Fig. R5**). At the same time, the iodine molecules lose electrons and form interhalogen. Of course, the halide electrolyte can only conduct lithium ions, but at the nanoscale, I₂/LIC7.2 can still achieve a high conversion rate and large capacity by **interface-mediated heterogeneous oxidation** reaction.

Secondly, to achieve the smallest possible particle size and maximize the nanointerfaces between I₂ and LIC7.2 in the I₂/LIC7.2 cathode, we employed a high-energy (500 rpm) and prolonged (12 h) ball milling process in a high-density tungsten carbide (WC) jar. This rigorous ball milling approach is widely used to produce nanoscale composite electrodes for all-solid-state batteries, creating extensive interfaces between

nanoparticles and thereby optimizing capacity release. For comparison, a hand-ground $I_2/LiClO_4$ electrode with inadequate contact interfaces was tested, resulting in a significantly lower capacity of just 23 mAh g^{-1} (Fig. R4c). This emphasizes the critical role of cathode preparation methods in achieving high conversion capacity for the I_2/I^+ reaction.

In the revised manuscript, we added the data and related discussion on the charging mechanism in line 203-236 on page 8-9.

Fig.R4 (a) High-resolution TEM image of the ball milled $I_2/LiClO_4$ composite. (b) EDX elemental mappings for C (red), In (green), Cl (yellow) and I (blue) of the ball milled $I_2/LiClO_4$ composite. (c) Direct charging profile of the hand ground $I_2/LiClO_4$ electrode.

Fig.R5 Schematic of the charging reaction mechanism of $I_2/LiClO_4$ cathode. During charging, I_2 loses electrons and the $LiClO_4$ loses Li ion, then I^+ combines with $LiClO_4$ to form interhalogen bonds at the interface.

5. Hu, R. *et al.* Dramatically enhanced reversibility of Li₂O in SnO₂-based electrodes: the effect of nanostructure on high initial reversible capacity. *Energy Environ. Sci.* **9**, 595–603 (2016).
6. Sun, Y. *et al.* High-capacity battery cathode prelithiation to offset initial lithium loss. *Nat. Energy* **1**, 15008 (2016).
7. Dong, W. *et al.* A Robust and Conductive Black Tin Oxide Nanostructure Makes Efficient Lithium-Ion Batteries Possible. *Adv. Mater.* **29**, 1700136 (2017).

5. Why are the Raman spectra at stage b and stage d in Figure 3b looking exactly the same? It is hard to believe even the noise looks the same as well, especially after the electrode has gone through discharging and charging.

Response: We apologize for this mistake and thank the reviewer for pointing it out. We wrongly used the same Raman data for stage b and stage d in Figure 3b because of similar file naming. The correct Raman figure has been added in the revised manuscript in Figure 3b in the revised manuscript and is also seen below in **Fig. R6**.

Fig.R6 Raman spectra of the I₂/LIC7.2 cathode at different SOC.

6. In Figure 5b, how was the coulombic efficiency calculated? Why would there be capacity degradation to 93% at cycle #200 when the average CE is claimed to be 100%?

Response: The coulombic efficiency is calculated by:

$$CE\% = \frac{\text{Charge capacity}}{\text{Discharge capacity}} * 100\%$$

and we take the average value for cycle 5 ~ 200 as the average CE. After carefully checking the data, the average CE should be 99.03%. We are sorry for this carelessness and have revised this in the manuscript.

Reviewer #2 (Remarks to the Author):

This work reports the application of high-capacity four-electron solid-conversion $I/I_2/I^+$ chemistry in solid-state lithium metal batteries, utilizing a chlorine-rich $Li_{4.2}InCl_{7.2}$ ionic conductive catholyte to facilitate I-Cl bond formation. The manuscript includes extensive characterization to elucidate the four-electron redox mechanism and conversion kinetics. However, some critical aspects, such as the phase evolution of $Li_{4.2}InCl_{7.2}$ upon I-Cl interhalogen bond formation and the solid-solid conversion efficiency, were not adequately addressed. Therefore, I cannot support its publication in its current form.

Response: We appreciate the reviewer for this valuable and constructive comment. Please see the point-to-point responses below.

1. Since $Li_{4.2}InCl_{7.2}$ is a Li^+ ion conductor with immobilized chloride ions in its crystal structure, the I_2/I^+ conversion efficiency and I-Cl bond formation likely depend on the contact between I_2 and Cl. How can effective contact be maintained while retaining a relatively high iodine content in the electrode?

Response: We thank the reviewer for raising this constructive comment.

Firstly, different with intercalation materials in which Li ion can easily diffuse, conversion electrodes usually show very poor ion diffusivity. Therefore, the electrochemical reaction efficiency of the conversion electrodes is highly related to the particle size of the active material, as well as the interface contact between active materials. The smaller the particle size and the greater the interface, the higher the conversion efficiency. **This interface-mediated heterogeneous oxidation mechanism has been reported and widely acknowledged for conversion-type metal oxide electrode materials of Li-ion battery**⁵⁻⁷. For example, Yi Cui et al. investigated the oxidation mechanism of Co/Li₂O composite⁶. They proved that when the material's size is at the micron scale, its charging capacity is quite limited (<100 mAh g⁻¹), and only at the nanoscale can it achieve a significantly large specific capacity during charging (over 600 mAh/g). During charging, nanoscale Co undergoes an **interface-mediated heterogeneous oxidation** induced by the interface with Li₂O. As Li₂O releases lithium ions, oxygen ions bind to the Co atoms on the surface of the metallic Co, with these Co atoms simultaneously losing electrons, forming Co₃O₄ (equation R1). Of course, both metallic Co and the charging product Co₃O₄ have poor ionic conductivity for oxygen ions. However, Co/Li₂O can still achieve a high charging capacity and a considerable amount of Co₃O₄ conversion. This is the result of interfacial-induced heterogeneous oxidation reactions at the nanoscale.

A similar reaction occurs in our study as shown in **Fig. R7**. Nanoscale iodine and lithium halide electrolyte also form a pair of charging reactants. During the charging process, the lithium halide electrolyte (LIC7.2) releases lithium ions, and the halide ions bind to the iodine on the surface of the iodine particles. At the same time, the iodine molecules lose electrons and form interhalogen (equation R2). Of course, the halide electrolyte can only conduct lithium ions, but at the nanoscale, I₂/LIC7.2 can still achieve a high conversion rate and large capacity by **interface-mediated heterogeneous oxidation** reaction.

Secondly, in our study, to achieve high conversion efficiency for the I₂/I⁺ redox reaction, we utilize a high-energy ball milling process (500 rpm for 12 hours) to prepare the I₂/LIC7.2 cathode. This intense milling leads to the formation of nanoparticles, creating a substantial number of interfaces between them. TEM results (**Fig. R8a**) confirm that both the I₂ and LIC7.2 domains are only a few nanometers in size. Additionally, EDX images (**Fig. R8b**) demonstrate homogeneous mixing of I₂/LIC7.2 at the nanoscale. This approach maximizes the capacity of I₂/I⁺ conversion in the I₂/LIC7.2 cathode. In contrast, the hand-ground I₂/LIC7.2 electrode shows a significantly lower capacity of only 23 mAh g⁻¹ (**Fig. R8c**), primarily due to larger particle sizes and insufficient contact interfaces.

Thirdly, we would like to point out that the weight ratio of I₂ is ~25wt% in I₂/LIC7.2 cathode. We find that when the I₂ weight ratio is 15 wt%, the I₂/LIC7.2 cathode shows similar specific capacity to the 25 wt% one. But when we further increase the I₂ weight ratio to 40 wt%, a very limited specific capacity of 20 mAh g⁻¹ is observed (**Fig. R8d**). This is because in the high weight ratio I₂/LIC7.2 cathode, I₂ inevitably forms large clusters. Inside the clusters, I₂ loses contact with LIC7.2 and thus there is very limited interface for I₂/I⁺ reaction to take place. In addition, it is difficult to form an effective conductive network with such low LIC7.2 catholyte content. This result again highlight the importance of the interface construction in achieving highly efficient I₂/I⁺ conversion.

We have added the related discussion on the charging mechanism in line 203-236 on page 8-9.

Fig.R7 Schematic of the charging reaction mechanism of I₂/LIC7.2 cathode. During charging, I₂ loses electrons and the LIC7.2 loses Li ion, then I⁺ combines with LIC7.2 to form interhalogen bonds at the interface.

Fig.R8 (a) High-resolution TEM image of the ball milled I₂/LIC7.2 composite. (b) EDX elemental mappings for C (red), In (green), Cl (yellow) and I (blue) of the ball milled I₂/LIC7.2 composite. (c) Direct charging profile of the hand ground I₂/LIC7.2 electrode. (d) Voltage profiles of I₂/LIC7.2-based ASS Li-I₂ batteries with different I₂ mass loading.

- Hu, R. *et al.* Dramatically enhanced reversibility of Li₂O in SnO₂-based electrodes: the effect of nanostructure on high initial reversible capacity. *Energy Environ. Sci.* **9**, 595–603 (2016).
- Sun, Y. *et al.* High-capacity battery cathode prelithiation to offset initial lithium loss. *Nat. Energy* **1**, 15008 (2016).
- Dong, W. *et al.* A Robust and Conductive Black Tin Oxide Nanostructure Makes Efficient Lithium-Ion Batteries Possible. *Adv. Mater.* **29**, 1700136 (2017).

2.Determining the phase evolution of Li_{4.2}InCl_{7.2} during charge/discharge cycles is necessary to understand the I-Cl interhalogen bond formation process. If the Li_{4.2}InCl_{7.2} phase is maintained during redox, as claimed in the manuscript, the electronegativity of chloride might decrease due to ionic or covalent bonding in the solid structure, potentially reducing its capacity to stabilize I⁺ species.

Response: We appreciate the reviewer's insightful comment. As suggested, we conducted XRD measurements on the I₂/LIC7.2 composite at various states of charge, and the results are presented in **Fig. R9**. Upon charging to 4 V, the LIC7.2 retains the same structure as its pristine state, with the exception of peak shifts to higher angles. This indicates that LIC7.2 maintains structural stability and

experiences lattice shrinkage due to delithiation during charging. Additionally, the structure remains well-preserved after discharge.

Based on the XRD results, we propose that during the charging process, LIC7.2 undergoes delithiation while retaining its original crystal structure. Simultaneously, I₂ molecules lose electrons to form I⁺, which binds with the surrounding delithiated LIC7.2 to form interhalogen bonds. During discharge, LIC7.2 becomes lithiated, leading to the breaking of interhalogen bonds and the reformation of I₂ molecules.

The XRD data have been included in the Supporting Information as Figure S13, and the corresponding discussion has been added to the revised manuscript in line 391-394 on page 15: “*Moreover, the LIC7.2 in the I₂/LIC7.2 composite cathode can also maintain excellent structural stability during cycling, as confirmed by XRD results (Figure S13). Note that a slight shift of the XRD peaks is observed for the charged and discharged samples, further proving a reversible lithiation/delithiation behavior of the LIC7.2*”

Fig.R9 XRD patterns of the I₂/LIC7.2 composite cathode at pristine, after charge and after discharge states. The minor peaks are attributed to the crystalline LIC6 solid electrolyte.

3. Two distinct charge plateaus are observed for the initial charge of I₂/LIC417 or I₂/LIC316 electrodes (Figure 2b and Figure S4). Does this indicate that the valence state of iodine in the I₂/LIC417 or I₂/LIC316 compounds is not 0? Please provide a detailed explanation.

Response: The reviewer is right. Due to the strong electron absorption capability of I₂, there is an electron transfer from KB to I₂ during high-energy ball milling process. This phenomenon was reported before by Jung et al⁸. As a result, the valence state of I₂ is not 0 in the pristine I₂/LIC7.2 (I₂/LIC6) electrode. The pristine ball milled I₂/LIC7.2 (I₂/LIC6) electrode shows a Raman shift at 162 cm⁻¹, a

characteristic peak of polyiodide I_5^- . The observed two charging plateaus for the $I_2/LIC7.2$ ($I_2/LIC6$) electrode corresponds to two different oxidation processes (**Fig. R10**). The higher charging plateau at 3.75 V is attributed to the oxidation from I_2 to I^+ . The lower charging plateau at 3.3 V with a capacity of 60 mAh g^{-1} is attributed to the conversion of reduced polyiodides to I_2 . Since the conversion from I_5^- to I_2 can only exhibit a capacity of 42.2 mAh g^{-1} , it is most likely that the polyiodides generated from the electron-withdrawn effect contains both I_5^- and small amount of polyiodides with lower valence state (such as I_3^-) which is not evident in the Raman spectra.

We have made related statement in the manuscript in line 123-128 on Page 5: “*The charging plateau at 3.3 V with a capacity around 60 mAh g^{-1} is attributed to the conversion of reduced polyiodides species to I_2 . During the ball milling process, I_2 , with its strong electron absorption capability, induces electron transfer from KB to I_2 and therefore leads to the formation of reduced polyiodides species such as I_5^- . As the conversion from I_5^- to I_2 has a theoretical capacity of 42.2 mAh g^{-1} , there are most likely a small amount of other further reduced polyiodides species (I_3^-) existing in the pristine cathode.*”

Fig.R10 Direct charging voltage profile of $I_2/LIC7.2$ cathode. Two plateaus are observed, with the lower one corresponding to the oxidation from reduced polyiodide species to I_2 , and the higher one corresponding to the oxidation from I_2 to I^+ .

8 Jung, N., Crowther, A. C., Kim, N., Kim, P. & Brus, L. Raman Enhancement on Graphene: Adsorbed and Intercalated Molecular Species. *ACS Nano* **4**, 7005–7013 (2010).

4. The manuscript mentions a Raman shift of I_2 at 167 cm^{-1} observed in pristine I_2/LYB and $I_2/LIC417$ cathodes (Figure 2c). However, this corresponds to the characteristic peak of I_5^- in other literature (J. Am. Chem. Soc. 2024, 146, 10, 6744–6752; Adv. Mater. 2022, 34, 2201716). Why is this the case? Additionally, a peak around $100\text{--}130 \text{ cm}^{-1}$ in the Raman spectrum, which changes during the redox reaction (Figure 2c and Figure 3b), should not be ignored.

Response: We feel sorry for this misleading and thank the reviewer for pointing this out. The Raman shift at 167 cm^{-1} for the pristine I_2/LYB and $\text{I}_2/\text{LIC7.2}$ cathodes indicates the formation of I_5^- , as the reduced I_2 species which absorbs electron transfer from KB during the ball milling process. We have made corresponding corrections in line 164-167 on page 6.

The peak around $100\sim 130\text{ cm}^{-1}$ appears in every Raman spectra for I_2 cathode at different SOC, but with varying intensities (**Fig. R11**). Therefore, we believe that this Raman shift is not attributed to any iodine species and most likely due to the background signal.

Fig.R11 Raman spectra of the commercial ICl compound, pristine/charged I_2/LYB and $\text{I}_2/\text{LIC7.2}$ cathodes.

5.The self-discharge performance of the four-electron Zn-I₂ full battery at different depths of discharge (DOD) should be provided.

Response: As suggested by the reviewer, we evaluated the self-discharged behavior of the four-electron Li-I₂ batteries at both OCV and fully-charged states (**Fig. R12**). A very stable voltage of 3.2 V could be maintained for 120 h after the battery assembly, which indicates there is no self-discharge. When the battery is charged to 4 V and then rests for 120 h, the voltage slowly drops to 3.75 V and then keeps stable. A high coulombic efficiency of 93.03% is achieved, higher than that of reported I_2 cathodes in liquid Zn-I₂ batteries⁹⁻¹¹, proving the advantage of the employment of I_2 cathode in the all-solid-state system.

We added this data and statement in the revised manuscript in line 418-423 on Page 16: “*This advantage is further demonstrated by the self-discharge performance. A very stable voltage of 3.2 V could be maintained for 120 h after the battery assembly, which indicates there is no self-discharge (Figure S16a). When the battery is charged to 4 V and then rests for 120 h, the voltage slowly drops to 3.75 V*

and then keeps stable (**Figure S16b**). The resulting coulombic efficiency of 93.03% surpasses that of reported I_2 cathodes in liquid Zn- I_2 batteries.”

Fig.R12 Self-discharge performance of the all-solid-state Li- I_2 battery at OCV and after charging.

9. Li, C. *et al.* Highly Reversible Zn Metal Anode Securing by Functional Electrolyte Modulation. *Adv. Energy Mater.* **n/a**, 2400872.
10. Zhang, P.-F. *et al.* Toward Shuttle-Free Zn- I_2 Battery: Anchoring and Catalyzing Iodine Conversion by High-Density P-Doping Sites in Carbon Host. *Adv. Funct. Mater.* **34**, 2306359 (2024).
11. Wang, T. *et al.* Surface Patterning of Metal Zinc Electrode with an In-Region Zincophilic Interface for High-Rate and Long-Cycle-Life Zinc Metal Anode. *Nano-Micro Lett.* **16**, 112 (2024).

6. XRD patterns after ten or twenty cycles, or for LIC417 in different states in I_2 /LIC417 cathodes, should be included.

Response: As suggested by the reviewer, we conducted XRD measurements for LIC after 100 cycles (**Fig. R13**) and LIC in different states in I_2 /LIC7.2 cathodes (**Fig. R9** above in comment 2). No decomposition product is observed for the LIC after 100 cycles. We added the related discussion and data in the revised manuscript in Figure 4 and Page 15.

Fig.R13 XRD patterns of the LIC7.2/KB cathode at pristine, after 1st discharge, after 1st charge and after 100 cycles.

7. In Figure 5c, the battery operates at 60 °C, exceeding the sublimation temperature of iodine. The impact of potential iodine phase changes on the battery's performance warrants discussion.

Response: We performed a TGA measurement for the I₂@KB composite, and the result is shown in Figure S15 and also below in **Fig. R14**. The iodine in the I₂@KB starts to sublime around 40°C and still keeps a weight percentage of 99% at 60°C. This could be attributed to the effective trapping of I₂ inside the pores of KB.

We added the related discussion in the revised manuscript in line 471-474 on Page 18: “It should be noted that bare I₂ easily sublimates at temperature higher than 40°C. The effective trapping of I₂ inside the pores of KB avoids I₂ loss at 60°C in the I₂/LIC7.2 composite cathode (**Figure S21**), and leads to the above good high-temperature performance.”

Fig.R14 TGA measurement of the ball milled I₂@KB composite.

8. Why was LIC316 chosen as the solid-state electrolyte rather than LIC417? Does Cl⁻ migrate from LIC417 to LIC316 under the concentration gradient?

Response: In our designed all-solid-state Li-I₂ batteries, LIC7.2 (ionic conductivity of 0.2 mS cm⁻¹) was used as catholyte to induce the I₂/I⁺ redox reaction, while crystalline LIC6 (over 1 mS cm⁻¹) with a higher conductivity was chosen as the solid-state electrolyte to provide fast ionic transport within the batteries. We added this discussion in the method part in Page 21.

As inorganic solid-state electrolytes, both LIC6 and LIC7.2 have a transfer number close to 1. Therefore, theoretically Cl⁻ is fixed inside the structures and unable to move. However, whether the Cl⁻ can migrate

under the concentration gradient and its effect on ionic transport still needs further experimental investigation.

Reviewer #3 (Remarks to the Author):

The manuscript prepared by Cheng et al. reports an all-solid-state Li-I₂ battery. The authors proposed that the battery operates based on I⁻/I₂/I⁺ (four-electron) redox. The authors reported detailed studies on the battery mechanism and cycling performance. Before the manuscript can be considered for publication, the following comments still need to be addressed:

Response: We thank the reviewer for these comments. Please see our point-to-point response below:

1. The specific capacity reported in this manuscript was based on the mass of I₂. According to the author, the LIC417 plays an important role in helping the battery achieve a higher capacity by interhalogen interaction between Cl⁻ and I⁺. Therefore, the mass of LIC417 should also be included in the calculation of specific capacity, not just the I₂ itself. The author should also report energy density calculated in this manner, in addition to specific capacity and areal capacity.

Response: We agree with the reviewer that the capacity based on the total mass of I₂ and LIC7.2 (Li_{4.2}InCl_{7.2}) should be included in the figures. We have specified the capacity calculation in every related figure caption and added the I₂+LIC7.2 based specific capacity to Fig. 5 in the revised manuscript now.

We would also like to point out that, although the lithium source is contributed by the LIC7.2 catholyte during I₂/I⁺ redox, we didn't intentionally increase the amount of catholyte in the I₂/LIC7.2 cathode to obtain four-electron conversion. The conversion-type cathode materials inherently require a relatively large amount of electrolyte, due to their poor ionic conductivities. For example, in all-solid-state Li-S batteries, a typical usage of the catholyte is 40-50 wt% in the composite cathode¹⁻⁴. In most systems, catholytes remain electrochemically inactive during cycling. However, the LIC7.2 catholyte used in the Li-I₂ system actively participates in the electrochemical process alongside I₂, while still maintaining its high ionic conductivity. We have simply maximized the capacity extraction based on this conventional amount of electrolyte, in order to enhance the capacity of the conversion reaction in all-solid-state electrodes.

1. Pan, H. et al. Carbon-free and binder-free Li-Al alloy anode enabling an all-solid-state Li-S battery with high energy and stability. *Sci. Adv.* **8**, eabn4372 (2022).
2. Hou, L.-P. et al. Improved interfacial electronic contacts powering high sulfur utilization in all-solid-state lithium-sulfur batteries. *Energy Storage Mater.* **25**, 436-442 (2020).
3. Liu, M. et al. Quantification of the Li-ion diffusion over an interface coating in all-solid-state batteries via NMR measurements. *Nat. Commun.* **12**, 5943 (2021).
4. Kim, J. T. et al. Manipulating Li₂S₂/Li₂S mixed discharge products of all-solid-state lithium sulfur batteries for improved cycle life. *Nat. Commun.* **14**, 6404 (2023).

2. The cycling data reported in Figure 5 has I₂ loading of 0.5 mg cm⁻². This is too low, and it is suggested that the author should report the 3.5 mg cm⁻² data in the main figure instead.

Response: As the reviewer suggested, we put the cycling data based on 3.5 mg cm^{-2} (**Fig. R15**) as Figure 5c and Figure 5d in the revised manuscript.

Fig. R15 (c) Rate performance of the battery with a mass loading of 3.5 mg cm^{-2} at room temperature. (d) Cycling stability with the corresponding coulombic efficiency of the battery at 1 C and room temperature. The I_2 mass loading is 3.5 mg cm^{-2} .

3. The rate capability of the battery when the loading is increased (3.5 mg cm^{-2} vs. 0.5 mg cm^{-2}) is dramatically decreased. The author should explain the reason for such decay in rate capability.

Response: Compared to intercalation cathodes, conversion-type active materials including S, O_2 , FeF_2 and I_2 often show poor ionic diffusion capability. As a result, they usually exhibit unsatisfactory rate performance at high mass loadings.

We added this discussion in the revised manuscript in line 446-448 on Page 17: “Noting that the rate capability at 3.5 mg cm^{-2} is decreased compared to that at a lower mass loading of 0.5 mg cm^{-2} . This is attributed to the low Li^+ diffusion coefficient in the conversion-type I_2 cathode.”

4. Can the Cl content be further increased in LIC417? Will this further improve the battery performance? Or is LIC417 the optimal composition?

Response: We thank the reviewer for raising this constructive comment. We synthesized the Li-In-Cl via ball milling with four different stoichiometry: Li_3InCl_6 (LIC6), $\text{Li}_{3.6}\text{InCl}_{6.6}$ (LIC6.6), $\text{Li}_{4.2}\text{InCl}_{7.2}$ (LIC7.2), and $\text{Li}_{4.8}\text{InCl}_{7.8}$ (LIC7.8). Their EIS results are shown in **Fig. R16a**. With more Cl content, the ionic conductivity drops from 0.63 mS/cm for LIC6 to 0.06 mS/cm for LIC7.8. We then prepared the I_2 composite cathode mixed with these different LIC catholytes and assemble them in all-solid-state (ASS) Li- I_2 cells, to evaluate their ability to activate I_2/I^+ redox couple. As shown in **Fig. R16b**, the charging capacities for $\text{I}_2/\text{LIC6}$, $\text{I}_2/\text{LIC6.6}$, $\text{I}_2/\text{LIC7.2}$ and $\text{I}_2/\text{LIC7.8}$ cathode are 120, 186, 261 and 267 mAh g^{-1} , respectively. Therefore, an enough high Cl content in LIC (≥ 7.2) is important to fully activate the I_2/I^+ redox reaction. To achieve both high ionic transport and full utilization of I_2/I^+ redox couple, the LIC7.2 is selected as the optimized catholyte in this system. We have added the related data and discussion in line 136-158 on page 5-6.

Fig.R16 and (a) Nyquist plots of the LIC6, LIC6.6, LIC7.2 and LIC7.8 at room temperature. (b) Voltage profiles of ASS Li-I₂ batteries based on different catholytes during the first charging at 0.05 C and at room temperature.

5. The XRD data present in Fig. 4f actually shows differences. Compared to the Pristine data, the After 1st discharge and After 1st charge XRD data have their XRD peaks slightly shifted to higher angles. Also, the peak at ~ 72° in pristine XRD data seems to disappear after the battery has cycled. The author should explain these differences. In addition, does the XRD pattern of LIC417 further changes when the battery has cycled for a while (ex. cycle 100)?

Response: The slight shift of the XRD peaks is most likely attributed to the lithiation/delithiation process during the discharge/charge process. Moreover, we also observe that the intensities of the XRD peak for the discharged/charged sample decrease a lot. This indicates that the redox process of LIC7.2 accompanies by a loss of crystallinity. As the reviewer suggested, we conducted XRD measurement for the LIC7.2 after 100 cycles (**Fig.R17**). No decomposition product is observed, indicating a superior long-term electrochemical stability of LIC7.2.

We added the related data in Figure 4 and discussion in line 388-394 on Page 15 in the revised manuscript: “However, unlike sulfide-based electrolytes, the redox reaction in LIC7.2 does not lead to its direct decomposition, as evidenced by the XRD patterns of the LIC7.2 cathode at pristine, charged, discharge and 100-cycled states shown in **Figure 4f**. Moreover, the LIC7.2 in the I₂/LIC7.2 composite cathode can also maintain excellent structural stability during cycling, as confirmed by XRD results (**Figure S13**). Note that a slight shift of the XRD peaks is observed for the charged and discharged samples, indicating a lithiation/delithiation behavior of the LIC7.2.”

Fig.R17 XRD patterns of the LIC7.2/KB cathode at pristine, after 1st discharge, after 1st charge and after 100 cycles.

6. In line 253, the sentence should refer to Fig. 3c, not Fig. 3d. Same for line 237, there is a mistake in figure referencing.

Response: We thank the reviewer for pointing out the mistake. We corrected the referencing in the revised manuscript.

Reviewer #4 (Remarks to the Author):

The authors demonstrate an all-solid-state Li-I2 battery that enables four-electron conversion chemistry (I-/I2/I+). Poor interfacial contact between porous cathode and solid-state electrolyte has generally been a significant issue, causing poor performance in all-solid-state batteries. In this work, the authors introduce a chlorine-rich lithium-ion conductor, LIC417 (Li4.2InCl7.2) as a catholyte. This addition to form I2/LIC417 composite cathode not only enhances Li-ion conduction pathway through improved interfacial contact but also promotes a second major redox chemistry (I2 <=> 2I-Cl) in addition to the commonly known 2I- <=> I2 conversion. This strategy contributes to enhanced capacity (almost twice as high as that of a conventional lithium-iodide battery) and improved cycle life at room temperature. To the best of our knowledge, this is the first demonstration of all-solid-state battery based on 4 electron interhalogen battery chemistries. The claims and conclusions made by the authors are all well supported by adequate datasets with thorough analysis and interpretation. We recommend this paper be published in Nature Communication after some minor revisions.

Response: We appreciate the reviewer for this positive evaluation of our work. Please see the point-to-point responses below.

1. In lines 146~148 of the main text, the authors states, "In the LIC417-based system, LIC417 is used as the catholyte, whereas the bulk solid-state electrolyte (SSE) remains LIC316 to ensure fast ionic transport within the battery". Figure 1 also depicts the battery cell structure, supporting this statement. However, a few pages later, the authors unexpectedly interpret their DRT data with the interfaces generated by LGPS (lines 273~275). They briefly mention that LIC316 and LGPS are used as a dual-layer electrolyte here, which is inconsistent with the earlier cell description and figure drawing presented in the manuscript. This could potentially confuse readers. Please revise cell-drawing in Figure 1 and the main text (lines 146~148) to accurately describe the Li-I2 battery cell studied in this paper. Please also include plausible reasons for choosing LIC316-LGPS bilayer over a single layer of LIC316 or LGPS (What is the main role of LGPS in this system? How would the cell behave if it was assembled with a single layered LIC316 SSE or LGPS SSE, in combination with I2/LIC416 composite cathode?)

Response: We feel sorry for this potential confusion. We have revised the figure drawing in Figure 1 and corresponding cell description.

As LIC316 is not stable against Li-In alloy, the sulfide LGPS is used as a protective layer here to avoid the reduction of LIC316. If only LGPS is adopted as the SSE, due to its low electrochemical oxidation limit (2.5 V), the cathode interface between LGPS and I₂/LIC417 would continuously deteriorate upon cycling. This sulfide/halide double-layer electrolyte configuration is very commonly seen in all-solid-state batteries^{12,13}.

We added the related discussion in the revised manuscript in the caption of Figure 1 in Page 4: *The crystalline LIC6 is used as the solid electrolyte layer due to its high ionic conductivity, and a layer of LGPS is adopted between LIC6 and Li-In anode to prevent the reduction of LIC6.*

12. Li, X. et al. Air-stable Li₃InCl₆ electrolyte with high voltage compatibility for all-solid-state batteries. *Energy Environ. Sci.* **12**, 2665–2671 (2019).

13. Li, X. et al. Water-Mediated Synthesis of a Superionic Halide Solid Electrolyte. *Angew. Chem. Int. Ed.* **58**, 16427–16432 (2019).

2. Please provide the thickness of the electrodes and solid-state electrolytes along with accurate cell stack/voice space information for those who might want to reproduce the data.

Response: We thank the reviewer for this comment. SEM measurements are conducted to evaluate the thickness and porosity information of the electrodes and solid-state electrolytes (**Fig. R18**). The thicknesses of the electrodes and solid-state electrolyte are 59 μm and 243 μm , respectively. The porosity inside the electrode and solid-state electrolyte are 7.44% and 3.71%, respectively. The cell stacking pressure is 50 MPa for the cycling tests. We added these information with data in Methods part in the revised manuscript.

Fig. R18 (a) Cross-section scanning electron microscopy image of the laminated I₂/LIC7.2 cathode (5.1 mg cm⁻²) and LIC6 solid electrolyte layer (50 mg). (b) Scanning electron microscopy image of the pressed LIC6 solid electrolyte layer. The porosity is calculated to be 3.71%. (c) Scanning electron microscopy image of the pressed I₂/LIC7.2 cathode. The porosity is calculated to be 7.44%.

3. The authors commented in lines 257-259 that the active I₂ mass in the I₂/LIC417 cathode is only about 25%, indicating room for improvement. To achieve higher I₂ loading beyond 25 wt%, it is crucial to supply sufficient chlorine to form the interhalogen charge product (I-Cl). However, increasing the iodine mass loading would reduce the relative mass loading of other cathode components including LIC417 catholyte, a source of chlorine. Therefore, there might be a theoretical limit to the possible proportion of I₂ in the cathode that enables the complete 4 electron conversion from the system. It would be very helpful if the authors could provide additional insights into this point based on their scientific understanding of the I-Cl forming mechanism and the interaction between LIC417 and I₂ (stoichiometric ratio?) in the main text. If the authors already have data showing the effect of increasing the proportion of iodine in the cathode (from 25 wt% to 30 ~ 50 wt%?), please include it in the supporting information. This would provide valuable information and guidance for further research in all-solid-state lithium-iodide battery development.

Response: We thank the reviewer for this valuable comment. As the reviewer noted, the I₂/LIC7.2 cathode has a theoretical limit for specific capacity and energy density. This limit exists because the I₂/I⁺ conversion activation strongly depends on the interhalogen interaction between I₂ and LIC7.2, with both components contributing to the cathode mass. We conducted the electrochemical tests on the I₂/LIC7.2 cathode with different I₂ mass loading (15 wt%, 25 wt% and 40 wt%). As shown in **Fig. R19** below, while the I₂/LIC7.2 with relatively low mass loadings can exhibit full capacity of the I₂/I⁺ conversion, the one with 40 wt% only shows very limited capacity. This limitation is due to the insufficient I-Cl coordination environment and the compromised ionic transport provided by the LIC7.2. To further improve the practical capacity of the I₂ cathode, it's crucial to develop advanced I₂ composite cathode structures with higher specific surface area and improved electronic/ionic conductivity. Simultaneously, investigating the I-Cl coordination mechanism in the solid-state system would aid in understanding the interplay between I₂ and the catholyte, which helps to optimize the usage of active I₂ and catholyte in the cathode and achieve higher energy density for the ASS Li-I₂ batteries.

We added the data and this discussion in the revised manuscript in line 507-516 on Page 19: “While the ASS Li-I₂ battery shows excellent electrochemical performance and high safety, it is worth noting that the active I₂ mass in the I₂/LIC7.2 cathode only accounted for ~25%. Further increasing the I₂ mass ratio to 40% leads to a very limited activation of the I₂/I⁺ redox, which is attributed to the insufficient I-Cl coordination environment near the I₂ particles, as well as a worse ionic conductivity. To further improve the practical capacity of the I₂ cathode, it's crucial to develop advanced I₂ composite cathode structures with higher specific surface area and improved electronic/ionic conductivity. Simultaneously, investigating the I-Cl coordination mechanism in the solid-state system would aid in understanding the interplay between I₂ and the catholyte, which helps to optimize the usage of active I₂ and catholyte in the cathode and achieve higher energy density for ASS Li-I₂ batteries.”

Fig. R19 Voltage profiles of I₂/LIC7.2-based ASS Li-I₂ batteries with different I₂ mass loading.

4. There are several typos and mislabelling errors in the figures

- In Figure 2 caption: (f) should be changed to (g) for Two dimensional charge density distribution.

- In line 198, please add (Figure 2f) at the end of the sentence ending with LYB.

- In line 199, Figure 2f should be changed to Figure 2g.

- In line 237, delete (Figure 3c)

- In line 253, Figure 3d should be changed to Figure 3c.

- In line 264, please add the full name of DRT

- In line 267, Figure 3e and 3f should be changed to Figure 3d and 3e

- In line 288, Figure 3g should be changed to Figure 3f.

Response: We greatly appreciate the reviewer for such careful review. We have corrected all the typos and figure mislabeling errors in the revised manuscript.

5. In line 422, please include the specific capacity value next to the areal capacity of 6.75 mAh cm⁻² for easier comparison with other systems.

Response: We thank the reviewer for this comment. Now the specific capacity value is included in the revised manuscript in line 468-469 on Page 17: “an areal capacity of 6.75 mAh cm⁻² (412 mAh g⁻¹ based on I₂ and 137.4 mAh g⁻¹ based on I₂/LIC7.2) is achieved at a current of 0.1 C.”

Reviewer #5 (Remarks to the Author):

Response: We thank the reviewer for the valuable review.

Response to Reviewers' Comments

We would like to thank all the reviewers for their constructive comments and suggestions, which has in our view significantly raised the quality of the manuscript (NCOMMS-24-31566A). We have modified the manuscript accordingly, and listed the detailed corrections below point by point for each reviewer. Moreover, all revised portion has been marked in yellow in the revised manuscript. The main corrections and the responses to the reviewers' comments are as follows.

Point-to-Point Responses (see next page)

Reviewer #1 (Remarks to the Author):

The authors have made major improvements on the manuscript. Most of my comments were addressed. There are a few more questions below to answer before this manuscript can be published.

Response: We appreciate the reviewer for this positive and valuable comment. Please see the point-to-point responses below.

1. The authors need to make the statement clear regarding the specific capacity in the beginning of the manuscript. "...Based on the $\Gamma/I_2/I^+$ four-electron chemistry, the as-designed all-solid-state Li-I₂ batteries deliver a high specific capacity of 449 mAh g⁻¹ based on I₂ mass and an impressive cycling stability over 600 cycles with a capacity retention of 93% at 1C and at room temperature..." To avoid confusion, the authors should add the rate at which the high specific capacity of 449 mAh g⁻¹ was achieved.

Response: We thank the reviewer for pointing out this confusing point. We now specify the rate behind the capacity to avoid possible confusion, as also seen below:

"Based on the $\Gamma/I_2/I^+$ four-electron chemistry, the as-designed all-solid-state Li-I₂ batteries deliver a high specific capacity of 449 mAh g⁻¹ at 0.1C based on I₂ mass and an impressive cycling stability over 600 cycles with a capacity retention of 93% at 1C and at room temperature".

2. The specific capacity based on the mass of I₂ and LIC should be labeled on Fig 2b for a fair comparison with Fig 2a.

Response: We agree. We added the specific capacity based on the mass of I₂ and catholyte in Fig 2b, as can be seen below:

Fig R1. Direct charging voltage profiles of the I₂/LYB, I₂/LIC6, I₂/LIC6.6, I₂/LIC7.2 and I₂/LIC7.8

cathode. The specific capacities based on the I₂ mass and the total mass of I₂ and catholyte are displayed in bottom and top x-axis, respectively.

3. The activation mechanism for I₂/I⁺ is still vague at this point. The mechanism of Co/Li₂O system does not really apply in this case. Co atoms bond strongly with O atoms after delithiation, but how did iodine atom bond with LIC? Is it covalent bond or ionic bond?

Response: We appreciate the reviewer for this valuable comment. In our opinion, there are two similarities between I₂/LIC7.2 system and Co/Li₂O system in terms of the charging mechanism. First, during charging, the delithiation and oxidation processes occur in two distinct phases in both systems, which we refer to as the 'heterogeneous oxidation' mechanism in this work. Second, the efficiency of this heterogeneous oxidation process is strongly influenced by the particle size and the interface contact conditions between these two phases.

We agree with the reviewer that the bond formation mechanism of I₂/LIC7.2 system is different from Co/Li₂O system. In the charged product of Co/Li₂O, Co atoms and O atoms are bonded in the form of ionic bonds. In the I₂/LIC7.2 system, during charging the I₂ loses electrons to form I⁺ and then bonds with Cl⁻ at the interface, in the form of covalent bonds. This is because I and Cl are both nonmetals and form a bond by sharing electron pairs. Moreover, the difference in electronegativity between I and Cl is relatively small (around 2.5 for I and 3.0 for Cl), which means the bond is somewhat polar, but this difference is not enough to create an ionic bond. The formation of covalent bonds between nonmetal elements within the same group or adjacent groups is a common phenomenon in materials science. For instance, oxygen and sulfur form sulfate groups (SO₄²⁻), while phosphorus and oxygen form phosphate groups (PO₄²⁻). These anionic groups, bonded covalently, can further bond with metal elements via ionic bonds to form widely used electrode or electrolyte materials. Examples include lithium sulfate (Li₂SO₄), lithium iron phosphate (LiFePO₄), and lithium aluminum germanium phosphate (Li_{1.5}Al_{0.5}Ge_{1.5}(PO₄)₃). A similar phenomenon occurs in the charge product of the I₂/LIC7.2 system, where Cl ions bond ionically with Li and In while forming covalent bonds with I at the interface.

We further performed Solid-state UV vis spectroscopy measurement to confirm the formation of ICl bonds in the charged I₂/LIC7.2. As seen in Fig. R2 below, pristine I₂/LIC7.2 shows clear iodine molecular adsorption peaks at 460 nm. In the UV vis spectrum of charged I₂/LIC7.2, an absorption peak appear at around 350 nm, in analogous to the previous reports¹, which is assigned to the formation of ICl interhalogens. **This finding aligns with our Raman and XPS results (Fig.R3)**, which also strongly support the presence of ICl interhalogens, further validating our conclusions. We have included the UV-Vis spectrum data in the supporting information and related discussion in line 172-174, page 7:

“This is consistent with the solid-state UV-Vis spectroscopy results (**Figure S5**), which show a strong absorption peak at around 350 nm—a characteristic peak of ICl interhalogen formation¹ for the charged I₂/LIC7.2.”

1. Zou, Y. et al. A four-electron Zn-I2 aqueous battery enabled by reversible I⁻/I²⁺ conversion. *Nat. Commun.* **12**, 170 (2021).

Fig R2. Solid-state UV vis spectrum of the I₂/LIC7.2 composite (a) at pristine and (b) after charging.

Fig R3. (a) Raman spectra of the commercial ICl compound, pristine/charged I₂/LYB and I₂/LIC7.2 cathodes. (b) I 3d XPS spectra of the pristine and charged I₂/LIC7.2 cathode.

To make the activation mechanism clearer, we further clarify the bond formation process in the revised manuscript in line 229-233, page 9: “It should be pointed out that the difference between Co/Li₂O and I₂/LIC7.2 systems is that, Co atoms ionically bond with O atoms after charging, while the activated I ions covalently bond with Cl ions. This is because 1) I and Cl are both nonmetals and form a bond by sharing electron pairs, 2) the different in electronegativity between I and Cl is not large enough to create an ionic bond”.

4. In Fig 5d, the I₂/LIC7.2 cathode is only able to deliver a specific capacity of 80 mAh/g with I₂ mass loading of 3.5 mg cm⁻² at 0.5C, if you consider the mass of I₂ and LIC7.2. How is authors' view on the practical use of such cathode in ASSB?

Response: We appreciate the reviewer's insightful comment. In our study, 50 wt% of LIC7.2 was incorporated into the I₂/LIC7.2 cathode, serving as both the ionic pathway and a partial lithium source for the I₂/I⁺ conversion. This catholyte amount is commonly used for conversion-based cathodes (S, Li₂S, FeF₂) in solid-state batteries due to their inherently low Li-ion diffusivity.

Reducing the catholyte content in the cathode composite is indeed necessary to achieve higher energy density in practical applications. However, this reduction can compromise Li-ion conductivity in the composite cathode and affect the efficiency of the I₂/I⁺ redox couple activation. To fully utilize the I₂/I⁺ redox couple, a minimum amount of LIC7.2 catholyte is required. Thus, the development of advanced I₂ composite cathode structures with higher specific surface area, as well as halide solid electrolytes with high lithium content, low molecular weight (e.g., Al or Mg as M element and F, O, or (and) Cl as X element in Li_yMX_z halide), high conductivity, and the ability to coordinate with I⁺ is essential for achieving practical, high-energy-density four-electron I₂ cathodes in all-solid-state batteries.

Reviewer #2 (Remarks to the Author):

The authors have emphasized the charge mechanism in the revised manuscript, stating that "during the charging process, the lithium halide electrolyte (LIC7.2) releases lithium ions, and the halide ions bind to the iodine on the surface of the iodine particles. At the same time, the iodine molecules lose electrons and form interhalogen bonds." However, it is clear that chloride ions remain ionically bonded in the delithiated catholyte phase, as confirmed by the XRD pattern (Fig.R9). Previous studies on the formation of the ICl interhalogen phase in battery systems have shown that this process depends on the electronegativity of chloride ions, with free chloride ions being more favorable for this formation. Therefore, it is crucial for the authors to provide further evidence demonstrating that I⁺ binds with the surrounding delithiated LIC7.2 to form interhalogen bonds. This is a critical point that directly impacts the validity of the findings presented in this work.

Response: We thank the reviewer for this comment. We agree that the free chloride ions are more favorable for forming ICl interhalogen bonds. In the I₂/LIC7.2 system, chloride ions initially bond with Li and In ions. During the charging process, delithiation of LIC7.2 breaks the Li-Cl bonds, leaving Cl with unpaired electrons that can subsequently bond with activated I⁺ ions. Due to the immobility of chloride ions, this process can only occur at the interface between I₂ particles and LIC7.2 particles.

As the reviewer suggested, we further performed Solid-state UV vis spectroscopy measurement to confirm the formation of ICl bonds in the charged I₂/LIC7.2. As seen in Fig. R2 below, pristine I₂/LIC7.2 shows clear iodine molecular adsorption peaks at 460 nm. In the UV vis spectrum of charged I₂/LIC7.2, an absorption peak appear at around 350 nm, in analogous to the previous reports¹, which is assigned to the formation of ICl interhalogens. **This finding aligns with our Raman and XPS results (Fig.R3),** which also strongly support the presence of ICl interhalogens, further validating our conclusions. We have included the UV-Vis spectrum data in the supporting information and related discussion in line 172-174, page 7:

“This is consistent with the solid-state UV-Vis spectroscopy results (Figure S5), which show a strong absorption peak at around 350 nm—a characteristic peak of ICl interhalogen formation¹ for the charged I₂/LIC7.2.”

Fig R2. Solid-state UV vis spectrum of the $I_2/LIC7.2$ composite (a) at pristine and (b) after charging.

Fig R3. (a) Raman spectra of the commercial ICl compound, pristine/charged I_2/LYB and $I_2/LIC7.2$ cathodes. (b) $I\ 3d$ XPS spectra of the pristine and charged $I_2/LIC7.2$ cathode.

1. Zou, Y. et al. A four-electron Zn-I₂ aqueous battery enabled by reversible I⁻/I₂/I⁺ conversion. *Nat. Commun.* **12**, 170 (2021).

The supplemented Fig. R12 in the response letter showed a rapid OCV decay after 120 hours. Although this was observed in a solid-state cell where both electrodes are solidified, the underlying reasons for this decay must be addressed. It is crucial to investigate potential causes, including solid-solid interface stability issues, ion transport limitations, or degradation of active materials, to offer a comprehensive understanding of this phenomenon.

Response: We are sorry for the misleading message conveyed by Fig. R12. In this figure the voltage and corresponding applied current are both displayed. As shown below in Fig. R4a, after the cell assembly, the cell was rested for 120 h, showing a very stable OCV over this resting period. After 120 h, the cell was discharged at a current of -35 μA . This is why the reviewer sees the rapid voltage decay

after 120 h. For clarifying, we distinguish the rest and discharge/charge period in the revised figure, as seen below:

Fig R4. Self-discharge performance of the all-solid-state Li-I₂ battery at (a) OCV and (b) after charging. In (a) the battery was rested for 120 h followed by a discharge process. In (b) the battery was first charged to 4 V and then rested for 120 h, followed by a discharge process. The CE is calculated to be 93.03% for the battery with a 120 h rest process between charge and discharge.

Reviewer #3 (Remarks to the Author):

The authors have addressed my previous comments well and revised their manuscript accordingly.

The manuscript looks good now and is suggested to be accepted for publication.

Response: We thank the reviewer for the positive comment to our work.

Reviewer #4 (Remarks to the Author):

The authors have thoroughly addressed all the concerns and comments raised by the reviewers, and I recommend this paper for publication in Nature Comm.

Response: We thank the reviewer for the positive comment to our work.

Reviewer #5 (Remarks to the Author):

Response: We thank the reviewer for the valuable review.

Response to Reviewers' Comments

We greatly appreciate both reviewers for their constructive comments and suggestions, which has in our view significantly raised the quality of the manuscript (NCOMMS-24-31566B). The responses to the reviewers' comments are as follows.

Point-to-Point Responses (see next page)

Reviewer #1 (Remarks to the Author):

The authors have addressed my comments. The manuscript can now be published.

Response: We thank the reviewer for the positive comment to our work.

Reviewer #2 (Remarks to the Author):

All of my comments are addressed now. I recommend this paper for publication in Nat. Commun.

Response: We thank the reviewer for the positive comment to our work.